# SELF-SUPERVISED GRID CELLS WITHOUT PATH INTEGRATION

## ABSTRACT

Grid cells, found in the medial Entorhinal Cortex, are known for their regular spatial firing patterns. These cells have been proposed as the neural solution to a range of computational tasks, from performing path integration, to serving as a metric for space. Their exact function, however, remains fiercely debated. In this work, we explore the consequences of demanding distance preservation over small spatial scales in networks subject to a capacity constraint. We consider two distinct self-supervised models, a feedforward network that learns to solve a purely spatial encoding task, and a recurrent network that solves the same problem during path integration. We find that this task leads to the emergence of highly grid cell-like representations in both networks. However, the recurrent network also features units with band-like representations. We subsequently prune velocity inputs to subsets of recurrent units, and find that their grid score is negatively correlated with path integration contribution. Thus, grid cells emerge without path integration in the feedforward network, and they appear substantially less important than band cells for path integration in the recurrent network. Our work provides a minimal model for learning grid-like spatial representations, and questions the role of grid cells as neural path integrators. Instead, it seems that distance preservation and high population capacity is a more likely candidate task for learning grid cells in artificial neural networks.

## 1 INTRODUCTION

Known for their striking hexagonal spatial firing fields, grid cells (Hafting et al., 2005) of the medial Entorhinal Cortex (mEC) are thought to underpin several navigational abilities. These include path integration (Hafting et al., 2005; McNaughton et al., 2006; Burak and Fiete, 2009; Gil et al., 2017), forming a neural metric for space (Moser and Moser, 2008; Ginosar et al., 2023), vector navigation (Bush et al., 2015), and supporting memory and inference (Mulders et al., 2021; Whittington et al., 2020). Given the range of different functions believed to be supported by grid cells, it is natural to investigate which of these tasks, if any, are actually performed by these enigmatic cells.

In the modeling literature, emphasis has been placed on grid cells as path integrators, with computational models establishing that grid cells are capable of doing path integration (Burak and Fiete, 2009). Recently, it has also been shown that grid-like representations *emerge* in neural networks trained to path integrate (Cueva and Wei, 2018; Banino et al., 2018; Sorscher et al., 2022; Whittington et al., 2020; Xu et al., 2022; Dorrell et al., 2022; Schaeffer et al., 2023), which has been taken as evidence for grid cells performing path integration. However, under interventional cell ablations grid units are as important for path integration as randomly selected units (Nayebi et al., 2021) while band-like units are significantly more important (Schøyen et al., 2023). Moreover, these models are typically complex, featuring different architectures and activation functions, interacting label cell types (e.g. simulated place cell-like targets (Sorscher et al., 2022)), multiple regularization terms, additional constraints (e.g. path invariance (Schaeffer et al., 2023) or conformal isometry (Xu et al., 2022)), and large cell counts. All of these works report grid-like representations, making it difficult to disentangle exactly what function grid cells serve in the various models.

In this work, we therefore propose a minimal model of grid cell function inspired by other recent models (Xu et al., 2022; Dorrell et al., 2022; Schaeffer et al., 2023), and use this model to approach the question of whether grid cells do path integration. Concretely, we are inspired by the notion

of grid cells serving as a metric for space, and consider an objective function that requires distance preservation over small spatial scales. In addition, we impose an L1 capacity constraint, which favors distributed representations that occupy a minimal portion of the state space. We train neural networks to minimize the proposed objective functions, and find that strikingly hexagonal grid-like spatial representations emerge using these two simple ingredients.

To explore whether grid cells do path integration in our model, we ablate path integration itself by training a feedforward (FF) network to minimize a purely spatial version of the proposed objective, alongside a recurrent neural network (RNN) tasked with implicit path integration. We find that the feedforward network learns grid representations on par with those of the path integrating RNN model. However, some RNN units display distinct band cell-like (Krupic et al., 2012) spatial responses, not seen in the FF network. When pruning velocity inputs to sampled subsets of units, we find that path integration contribution is inversely correlated with the mean sample grid score, with band-type cells providing the largest contribution.

Our findings suggest that grid cells may serve as a distributed high-capacity, distance-preserving representation. However, grid cells do not appear to be defined by the task of path integration. On the contrary, grid cells appear to be relatively unimportant for path integration, suggesting that grid cells may be more suitable for defining neural metrics for space, at least in artificial neural networks.

## 2 RESULTS & DISCUSSION

### LOSS FUNCTION AND LEARNED REPRESENTATIONS

We consider the problem of training a representation that preserves distances in a neighborhood around a current location, as illustrated in Fig. 1a). Considering Cartesian coordinates $\mathbf{x}_t$ (e.g. along a trajectory) where $t$ indexes time, we propose the following objective

$$\mathcal{L} = \alpha \mathbb{E}_{t,t'} \left[ e^{-\frac{1}{2\sigma^2} \|\mathbf{x}_t - \mathbf{x}_{t'}\|^2} \left( \|\mathbf{x}_t - \mathbf{x}_{t'}\| - \|\mathbf{g}_t - \mathbf{g}_{t'}\| \right)^2 \right] + (1 - \alpha) \mathbb{E}[l_{cap}(\mathbf{g}_t)], \qquad (1)$$

where $\sigma$ is the envelope scale parameter that determines the width of the neighborhood distance preservation in the exponential term. This creates a window around each spatial location where the difference between physical and neural distances should be minimized. While this requirement is similar to demanding a conformal isometry (Xu et al., 2022), there are subtle, but behaviorally relevant differences (see Appendix A.9).

Gaussian radial basis functions are widely used and have been previously applied in e.g. normative models of grid cells to promote local separation of neural representations, as seen in Dorrell et al. (2022) and Schaeffer et al. (2023). $\alpha$ is a hyper-parameter to weight the different loss terms, and $\|\cdot\|$ denotes the Euclidean norm. $\mathbf{g}_t$ is the representation we wish to learn, which we parametrize with either a feedforward neural network or a recurrent neural network, as illustrated in Fig. 1c) and described in section 3.2. Although both models must solve the same spatial encoding task, the FF model takes direct Cartesian coordinate inputs, while the RNN only receives an initial Cartesian position and subsequently receives Cartesian velocities. To correctly encode subsequent positions and distances, the RNN therefore also needs to learn to path integrate. Notably, we constrain $\mathbf{g}_t$ to be non-negative and of constant L2 norm, i.e., $g_{it} \geq 0$ and $\|\mathbf{g}_t\| = 1$ for all $i, t$, similar to Xu et al. (2022) and Schaeffer et al. (2023). The first loss term is minimized when Euclidean distances in the learned representation $\mathbf{g}$ equal the target Euclidean distances in a neighborhood around the current location. The second loss term, $l_{cap}$, is a capacity term. Xu et al. (2022) and Schaeffer et al. (2023) posit that capacity constraints are conducive to grid-like representations, and Schaeffer et al. (2023) proposed an L2 activity-based regularization term to maximize representational capacity. We, on the other hand, propose using an L1 capacity term given by

$$l_{cap}(\mathbf{g}_t) = -\sum_i g_{it}. \qquad (2)$$

Using an L1 capacity term is markedly different from an L2 capacity term both empirically and mechanistically. An empirical investigation into the heterogeneous effects of using L2 capacity regularization instead of L1 is presented in Appendix A.8. A geometric illustration of when L1 capacity is optimal is illustrated in Fig. 1b). When $\mathbf{g}$ has constant L2 norm and non-negative elements, L2 capacity promotes representations with similar angles, but any angle (in the positive quadrant) is

equally rewarded. Akin to L2 capacity constraints, the L1 capacity constraint (2) will also promote representations with similar directions. However, the L1 capacity constraint encourages maximally distributed and correlated cell activities. In other words, the full population vector state space is ideally placed near the diagonal vector, with all units coactive.

Surprisingly, both FF and RNN models learn highly grid-like representations, as seen in the ratemaps in Fig. 1c), and quantified by grid scores in Fig. 1d). We further find in Fig. 1d) that their phase distribution is seemingly random and uniform within the unit cell of the grid pattern. The histogram of grid spacings is unimodally peaked, indicating a single module in both models. However, the orientation histogram for the FF model is bimodal, suggesting two modules, but with identical spacing. In Appendix Fig. A1a) and b), we perform a parameter sweep across $\alpha$ and $\sigma$, evaluating grid score and grid spacing, and demonstrating how grid spacing can be tuned by adjusting these parameters. Moreover, Appendix Fig. A9 shows that grid spacing and field size vary independently and can also be controlled using a third hyperparameter, $\rho$. Extending our model to include multiple modules could be achieved by partitioning it, as in Xu et al. (2022), and assigning different distance-preservation hyperparameters to each module. However, we consider this beyond the scope of the current study, as our focus is on analyzing the functional role of the emergent cell types while minimizing potential confounding factors. Finally, Fig. 1e) shows the population vectors projected to three dimensions using UMAP alongside persistence diagrams quantifying the number of persistent 1,2 and 3-dimensional cocycles. Both models show one 0D, two 1D, and one 2D hole, indicating a toroidal manifold, which is also evident in the accompanying UMAP projection, consistent with recent experimental evidence in biological grid cells (Gardner et al., 2022).

While both networks learn similar representations, we find that the RNN learned a small set of cells with band-like representations, which is also visible in Fig. 1d) as a small bump near zero-grid score (see Appendix A.6 for ratemaps of all units, for both networks). Why would the model add this extra band-like subpopulation to the RNN? One key difference between the FF and the RNN is that the RNN is required to path integrate. It therefore seems especially strange that the amount of grid-like cells is reduced when the network is also required to path integrate, considering that mechanistic theories advocate grid cells as the neural substrate for path integration, and normative models argue that grid cells appear in RNNs trained for path integration. Additionally, when evaluating the loss terms on distinct subpopulations, we observe in Fig. A1b) that units with high grid scores achieve a somewhat lower distance loss, while band-type cells afford slight improvements in capacity. This supports the notion that grid cells are optimal for distance preservation. To rule out that architectural differences induce the difference in observed patterns between model types, we also train RNNs without velocity input, and find that band patterns vanish (see Appendix A.5). We also find that band-type units become more prolific in networks trained in a high-speed setting (necessitating stronger path integration).

## 2.1 PRUNING & PATH INTEGRATION ABILITY

To investigate the role of different cell types during path integration, we selectively pruned velocity inputs to different cells as illustrated in Fig. 2a). By pruning velocity inputs, rather than network units directly (i.e. network states), we minimize off-target effects, as pattern formation and network stability should only be affected in cases where units require velocity input to perform state updates (i.e., path integrate). As an alternative to this approach, we also train a one-step linear decoder (see Appendix A7) to demonstrate that the network is in fact path integrating.

We categorized cell types with a grid score of less than 0.15 as band-like and the rest grid-like, as seen in Fig. 2b). Pruning velocity input to the band-like cells induces a stark increase in (path integration) error, as given by Eq. 4, over time, as shown in Fig. 2c). Comparably, pruning velocity input to high grid score subpopulations showed almost no change in error over time. Furthermore, we find that path integration performance is not strongly affected even when input to all grid units in the network is pruned (see Fig. A2). Figure 2e) displays the corresponding error when pruning uniformly across any cell (including both band and grid-like), which is accompanied by higher error. We hypothesise that this is due to the fact that low grid score units can be included in the subpopulation. We also find that path integration ability correlates with band-like tuning, during training (see Appendix A7).

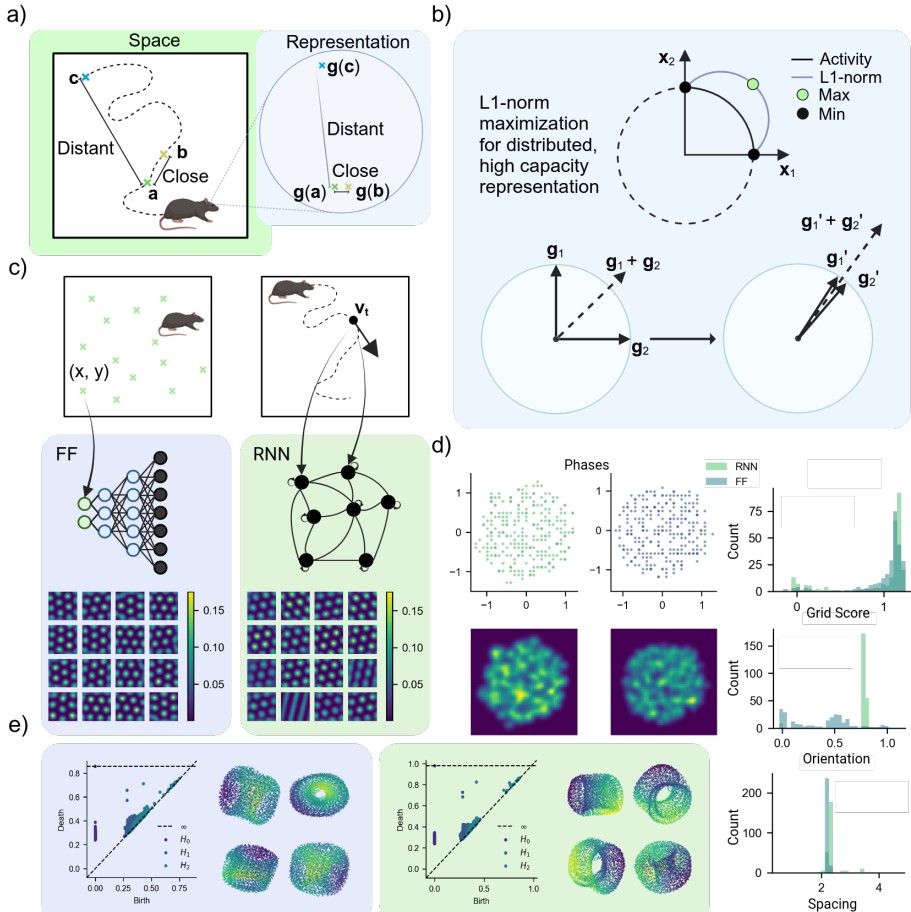

Figure 1: **Overview of models and objective function**. a) Illustration of the objective function: distant locations should be represented by distant population vector, close locations by close population vectors. b) Illustration of the L1 capacity constraint. c) Illustration of the investigated neural network architectures (feedforward and recurrent) and inputs. Below, ratemaps of a random selection of units are inset. d) Distributions of phases, grid scores, orientations, and spacing for both models. Orientations are given in radians, spacing is relative to environment dimensions (a $4\pi \times 4\pi$ square arena). e) Persistence diagram and 3D UMAP projection of population ratemaps, for feedforward and recurrent networks. For the feedforward network, results are shown for units with orientation in the range $[0.4, 0.8]$.

Schøyen et al. found similar results when pruning band-like cells in Sorscher et al.'s RNN model of grid cells. We further demonstrate in Fig. 2d) how the error is linearly related to grid score, where pruning low grid score units has a high impact, and conversely, pruning high grid score units yields low error. Finally, we also compare the initial state difference (ISD), as described in Equation 5, to later states along different trajectories when pruning in Fig. 2f). We see the ISD rise fastest with no pruning, as would be expected when moving away from the initial state. When pruning high grid score units, the trend is similar to no pruning, i.e., the neural representation is moved away from its initial state over time, as one would expect if the network was still path integrating. However, when pruning low grid score units (band cells), we see a much flatter ISD, indicating that the neural state does not change as much. In other words, the neural path integrator is close to being turned off. This provides compelling evidence that band cells, not grid cells, do path integration, at least in recurrently connected artificial neural network models.

In other recent work that incorporates path integration in the learning task, most of the learned spatial representations appear grid-like Xu et al. (2022); Schaeffer et al. (2023). Thus, in these models, it

appears that grid cells alone are responsible for the imposed objectives, including path integration. Moreover, ablating each term of their losses provides compelling evidence for the need of each component for robust pattern generation. However, Schaeffer et al. (2023) report band-like tunings for some runs, and compared to our model, features a velocity-dependent, MLP recurrent weight matrix that may obscure the contribution of other cell types (see Appendix A.4 for a detailed model comparison). In Xu et al. (2022), while a high proportion of cells were classified as grid cells, other types, such as band-like cells, can be observed in some reported LSTM units.

Importantly, neither model has explored whether path integration is a necessary condition for grid pattern emergence. While this would be an intriguing test, it is unclear how these models could be adapted to non-path-integration domains, as certain features they identify as essential for pattern formation—such as path invariance and trajectory permutations—rely on path integration. For a comprehensive comparison between our model and others, including Xu et al. (2024); Dorrell et al. (2022); Sorscher et al. (2022); Xu et al. (2022); Schaeffer et al. (2023), see Appendix A.4.

## 2.2 Pattern formation, connectivity, and generalizability

To investigate whether all hexagonal patterns are created equal, we examined the pattern formation, connectivity structure, and generalization capabilities of the two models. In the bottom row of Fig. 3a), a sorted subset of feedforward unit ratemaps is shown, scaled by their outgoing weights to a selected output unit (displayed as a large ratemap to the right), as detailed in section 3.5. For the FF model, ratemaps of the penultimate layer serve as the basis representations forming the grid pattern in the final layer. This basis representation is diverse, encompassing various patterns such as place-like (see the final three ratemaps) and single-band-like (see ratemap in the final row, second column) ratemaps. These basis representations are non-periodic, indicating that the corresponding downstream grid representations cannot maintain periodicity outside their training domain. This non-periodic nature can also be seen directly in the FF-network architecture, which uses non-periodic activation functions. The bottom row of Fig. 3c) confirms that the network does not generalize the grid pattern outside its training domain.

The RNN, while facing a similar out-of-domain initialization challenge as the FF network, can potentially generalize beyond the training arena boundaries using a learned, periodic path integrator circuit. Subsequent ratemaps and their cumulative pattern formation rely on previously similar periodic patterns, as observed in the top rows of Fig. 3a). Fig. 3c) demonstrates that initializing the RNN inside its training domain and allowing it to path integrate along long sequences beyond the domain reveals a clear periodic generalization. Conversely, starting outside its training domain results in the same non-generalization behavior as the FF network. Additionally, Fig. 3b) shows that the connectivity profile of the recurrent cells follows a short-range excitation and long-range inhibition principle with respect to cells with neighboring phases, which is a known structure for generating grid-like representations (Burak and Fiete, 2009). Interestingly, the band-like cells exhibit an excitation-inhibition connectivity profile shifted by their phase. Combined with the previous finding that band cells function as the neural path integrator circuit in the RNN, this suggests a mechanism for path integration through an excitation-inhibition shift of population activity in the wave direction of the band. Fig. 3b) also shows that the input weight matrix has a particular structure: velocities are projected along band directions (forming a hexagonal pattern in connectivity). Furthermore, velocity weights tend to decrease with increasing grid score, possibly suggesting that grid cells are less tuned towards integrating velocities.

## 2.3 Summary and outlook

Our approach, featuring a minimal model (only two loss terms and a model without path integration), has allowed us to isolate key factors contributing to the emergence of grid-like representations. This simplification contrasts with previous complex models, which, while effective in generating grid-like patterns, often obscure the specific contributions of grid cells to navigational tasks due to their many-sided nature. Our findings demonstrate that grid cells, as encoded in feed-forward and recurrent neural networks, can emerge when optimized to preserve local spatial distances under a capacity constraint. This aligns with previous theories positing that grid cells contribute to spatial navigation by providing a consistent metric for space (Moser and Moser, 2008; Stemmler et al., 2015; Ginosar et al., 2023; Schøyen et al., 2024).

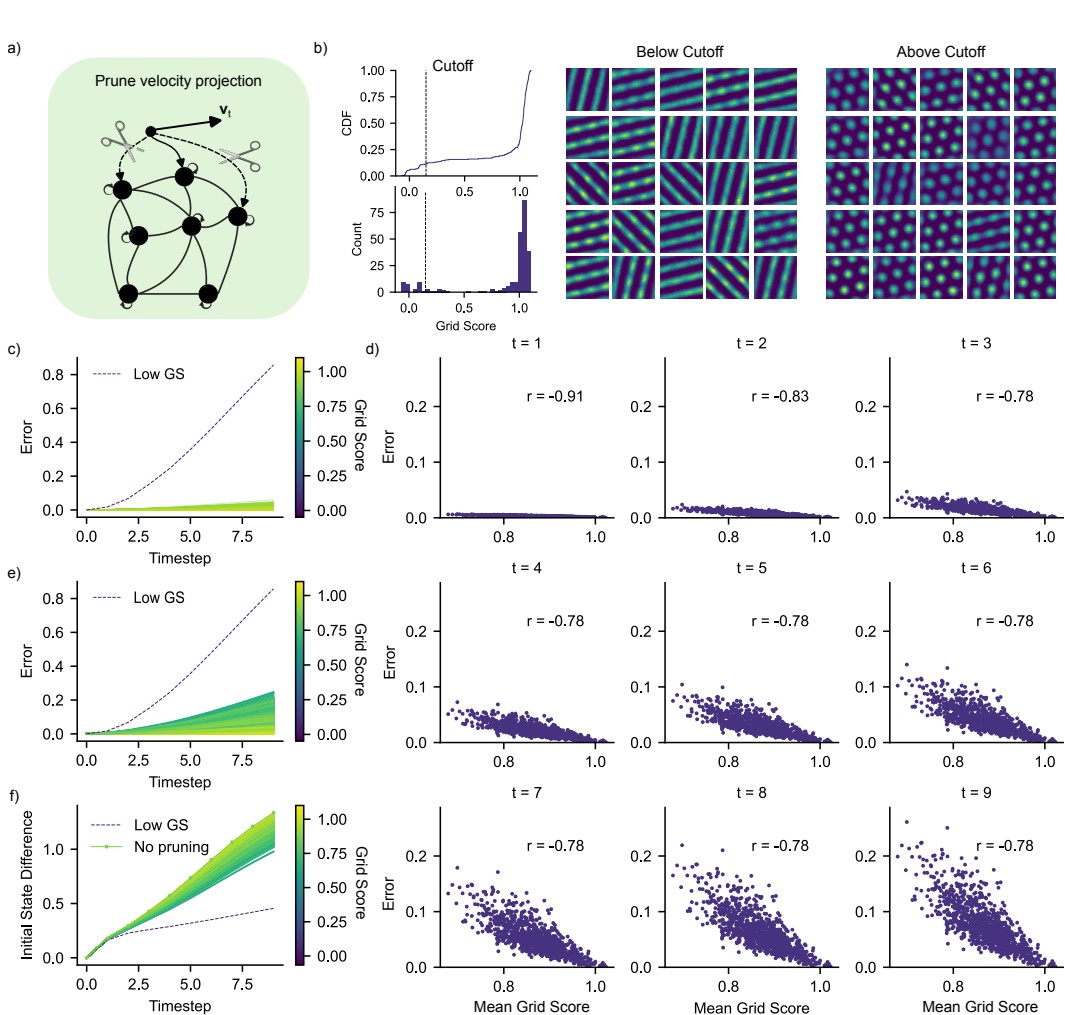

Figure 2: **Results of pruning velocity inputs to the RNN**. a) Illustration of the velocity input pruning: during pruning, velocity projections to randomly selected subsets of units are silenced. b) Grid score distribution, with indicated cutoff for low grid score units. Shown are also ratemaps for low and high grid score units. c) Path integration error for pruning of 1000 subpopulations among units with high grid score. Also shown is the error for low grid score units (dashed). d) Path integration error at each trajectory timestep, for pruning of randomly sampled subpopulations of units. Inset is the Pearson correlation coefficient between subpopulation mean grid score and the PI error. e) As in c), for random subsamplings of the full population. f) Average initial state distance, for pruning random subpopulations of the full population. Also shown is the initial state distance for the low grid score selection (dashed).

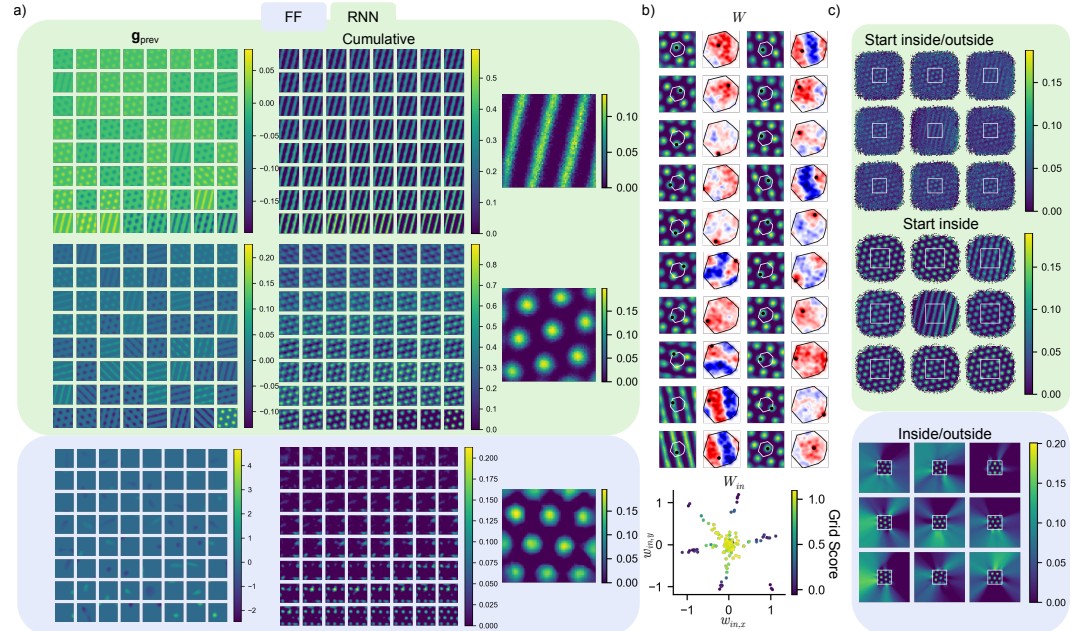

Figure 3: **Pattern formation and out-of-bounds behaviors**. a) Example pattern formation for select recurrent units (top and middle), and a feedforward unit (bottom); ratemaps inset on the right. $\mathbf{g}_{prev}$ denotes previous activations (penultimate layer activation for FF, second-to-last state for RNN), while cumulative representations shows the cumulative sum of weighted unit inputs. b) Top: Ratemaps, alongside spatial connectivity of the recurrent network. Shown is the outgoing recurrent weight, as a function of unit spatial phase. Also inset is the convex hull of all ensemble phases. Red indicates excitation (positive weight), blue inhibition (negative weight). Bottom: x- and y- components of the input matrix of the RNN, shaded by grid score of target unit. c) Evaluation of recurrent (top) and feedforward units (bottom) units when the network is the environment is extended beyond the original training regime (white square). For the RNN, representations are shown for trajectories starting within the training enclosure, and both inside and outside.

Our results challenge the widely-held view that grid cells are integral to path integration. In our experiments, feed-forward networks, which lack an explicit path integration mechanism, still developed grid-like representations comparable to those in the path-integrating recurrent network. This observation suggests that the emergence of grid cells is not contingent on the process of path integration. Instead, it indicates that grid cells may function to encode spatial relationships.

Pruning velocity inputs further allowed us to disentangle the contributions of different cell types to path integration. By comparing the representations of a pruned network to a non-pruned network, we could directly ascertain that path integration was not critically dependent on grid cells, but seemingly rather on band-type units, echoing other recent findings (Schøyen et al., 2023). The inverse correlation between the contribution to path integration and the mean sample grid score further reinforces the notion that grid cells might not be as crucial for path integration as previously thought.

In summary, our findings suggests a reevaluation of the role of grid cells within the neural navigation framework. While grid cells clearly provide a powerful spatial metric, their necessity for path integration is less obvious. This insight not only advances our understanding of grid cell functionality but also prompts a reconsideration of how spatial representations are constructed and utilized in both biological and artificial systems. Future research should continue to explore the interplay between different cell types in the entorhinal cortex, as well as investigate how these findings can inform the development of more efficient and interpretable models of spatial navigation.

## 2.4 LIMITATIONS

While our work provides evidence supporting the role of grid cells as a high capacity distance preserving representation, our model operates in a simplified domain, with Cartesian coordinate and velocity inputs. One could therefore consider extending the recurrent model to include e.g. more expressive input projections (such as Schaeffer et al. (2023)), or use other self-motion information, such as simulated head direction cell (Taube et al., 1990) input.

From a biological perspective, the use of label distances during training is also implausible. A related limitation is the use of Euclidean distances in computing the loss function. While this might be sufficient in an open arena such the one used in this work, more naturalistic settings, with e.g. interior walls may require other distance functions to capture the observed deformation of grid patterns in e.g. exotic geometries (Ginosar et al., 2023). Using the Euclidean distance between state vectors may also lead to inaccuracies when computing the loss. As an alternative, one could compute state distances using the metric induced by the neural network model. However, when comparing over smaller distances (which are implicitly enforced by $\sigma$), the Euclidean distance function could still be a fair approximation.

## 3 METHODS

### 3.1 DATA & INPUT TO NETWORKS

We considered two distinct neural networks in this work, with different inputs. The feedforward network received batches of Cartesian coordinate inputs $(x, y)$, sampled randomly and uniformly from a square region with side lengths $4\pi \times 4\pi$ (arbitrary units). The recurrent network, on the other hand, received Cartesian velocity inputs $(v_x, v_y)$, along trajectories sampled in the same square region. Additionally, the recurrent network was given the starting location $(x_0, y_0)$ of each trajectory (in Cartesian coordinates), to form a suitable initial state.

To generate trajectories, a bouncing procedure was used. Starting locations were sampled randomly and uniformly within the square arena. At each step, head directions were sampled according to a von Mises distribution with scale $\kappa = 4\pi$, and step sizes drawn from a Rayleigh distribution with scale parameter $s = 0.15$. Subsequently, we checked whether the resulting step landed outside the enclosure. If so, the component of the velocity vector normal to the offending boundary was reversed, effectively causing an elastic collision with the wall, keeping the trajectory inside. Otherwise, the procedure was repeated until the desired number of timesteps and trajectories was achieved. Due to the simplicity of the data, no datasets were pre-created, and all training data was new to the networks, i.e. created on the fly.

### 3.2 NEURAL NETWORKS & TRAINING

We consider two distinct architectures: A fully connected feedforward network and a recurrent neural network. The FF model consisted of two hidden layers, with 64 and 128 units, respectively, followed by an output layer of size $n_g = 256$ units, transforming 2D Cartesian coordinates to a latent space of $n_g$ dimensions. We applied the ReLU activation function after each hidden layer, and normReLU after the output layer. normReLU is a normalized ReLU function, which we take to be given by normReLU$(\mathbf{x}) = \text{ReLU}(\mathbf{x})/\text{maximum}(||\text{ReLU}(\mathbf{x})||, \varepsilon)$, with $\varepsilon = 10^{-12}$ a small constant to avoid zero division. Finally, we initialize the weights of each layer in the FF model uniformly between $-\sqrt{k}$ and $\sqrt{k}$, where $k$ is the number of input features to the layer.

Similar to the feedforward model, the RNN model featured $n_g = 256$ recurrently connected units. The state of the RNN model at a time $t$ was given by

$$\mathbf{g}_t = \text{normReLU}(W\mathbf{g}_{t-1} + W_{in}\mathbf{v}_t) \tag{3}$$

where $W \in \mathbb{R}^{n_g \times n_g}$ is a matrix of recurrent weights, $W_{in} \in \mathbb{R}^{n_g \times 2}$ a matrix of input weights, while $\mathbf{v}_t$ the velocity input at $t$. The initial state of the RNN was encoded with a feedforward neural network, with the exact same architecture as the FF model. In this case, the initial Cartesian position of the intended trajectory was provided as input. The weights in the recurrent matrix $W$ were initialized to the identity to mitigate vanishing/exploding gradients, similar to Le et al. (2015),

while the weights of the input matrix $W_{in}$ were initialized uniformly, similarly to the FF model. We also initialized the RNN according to a uniform distribution (Fig. A3), with similar results, but slower convergence times. We therefore use the identity initialization throughout this work.

Both models were trained with a mini-batch size of 64 using the Adam optimizer with a learning rate of $10^{-3}$ (Kingma and Ba, 2017). We set $n_g = 256$ for both models, and trained the RNN on 10-timestep trajectories. The RNN was trained for a total of 50000 training steps and the FF network for 100000 steps, on unseen data. Note that the RNN initial state encoder was trained from scratch when training the RNN, and did not reuse the previous, independent FF model.

The final parameters needed for training are $\sigma$ and $\alpha$, which define the loss function (1). $\sigma$ is an envelope scale parameter for the distance preservation loss term (See Appendix A.9 for more). $\alpha$ determines the relative weighting of the two terms in the loss, where $\alpha$ is the weight for the distance loss term and $1 - \alpha$ is the weight for the capacity loss term, chosen such that $\alpha \in [0, 1]$. We performed a grid search for $\alpha$ and $\sigma$ for both models to explore the impact of this weighting on the grid score and scale of the representations. The values for $\alpha$ were chosen uniformly between 0.01 and 0.99, while $\sigma$ values were selected empirically around values that were seen to yield models high grid score during initial testing. After the grid search (Fig. A1 a)), the values $\sigma = 1.2$ and $\alpha = 0.54$ were chosen for the FF and RNN models as this yielded high grid scores for both models, even though multiple other combinations of $\alpha$ and $\sigma$ gave similar results. See Appendix A.1 and A.6 for more on the effects of $\alpha$ on learned representations.

### 3.3 VELOCITY PRUNING

To investigate the influence of different RNN unit types on path integration ability, we ablated velocity inputs to subsets of recurrent units. Specifically, the velocity-pruned recurrent state was given by

$$\tilde{\mathbf{g}}_t = \text{normReLU}(W\mathbf{g}_{t-1} + \mathbf{m} \odot (W_{in}\mathbf{v}_t)),$$

where $\mathbf{m}$ is a binary mask that silences velocity input to select units, and $\odot$ the Hadamard product. RNN units were subselected so that a given subpopulation featured as many units as the number of low-gridscore units ($n = 29$; grid score cutoff 0.15). The subselection procedure was done for the full ensemble (random), as well as high grid-score units (grid score above cutoff). In each case, 1000 subpopulations were sampled randomly, with equal probability across units.

Since the proposed objective does not feature any decoding into a known target representation such as Euclidean coordinates, an alternate method of assessing whether the network is path integrating correctly is needed. We therefore computed the mean square error between velocity-pruned representations and a target representation created by running the network on the same velocity input without pruning. In other words,

$$\text{Error}_t = \frac{1}{M} \sum_{i=1}^{M} ||\mathbf{g}_t^i - \tilde{\mathbf{g}}_t^i||^2, \tag{4}$$

where $M = 10000$ is the number of evaluated trajectories, and $T = 10$ the trajectory length. As a baseline, we also computed the mean squared difference with the initial RNN state, i.e.

$$\text{ISD}_t = \frac{1}{M} \sum_{i=1}^{M} ||\mathbf{g}_0^i - \tilde{\mathbf{g}}_t^i||^2, \tag{5}$$

for velocity pruning to randomly sampled subpopulations, as described previously. To further establish that the network is capable of path integration, we also explore a trainable, one-step linear decoder (See. Appendix A.7).

### 3.4 GRID STATISTICS

To quantify the properties of the learned representations, we compute the grid score, orientation, spacing and phase of smoothed unit ratemaps. Smoothing was done with a Gaussian kernel, using the Astropy library (The Astropy Collaboration, 2022), with a standard deviation of 2 pixels, for $64 \times 64$ ratemaps of unit activity. Grid scores were computed as the difference between the smallest and largest correlations, when comparing autocorrelogram annuli correlations at 60 and 30 degrees,

respectively. Orientations were computed as the smallest angle of the six innermost peaks of the autocorrelogram (excluding the origin) with the horizontal. The grid spacing was computed as the average distance to the six innermost peaks. Finally, grid phases were taken to be the displacement of the closest grid peak to the origin of the ratemap.

### 3.5 PATTERN FORMATION

To better understand how representations emerge in the feedforward and recurrent networks, we visualized the pattern formation process for select units. To do so, we weighted the activity of input ratemaps to a given unit by the network weight relating them. Then, weighted inputs were sorted according to their L2 norm over all spatial bins, and pattern forming was visualized using the cumulative sum of all weighted input ratemaps.

### 3.6 OUT-OF-BOUNDS EVALUATION

To explore the ability of the networks to generalize beyond their training regime, we evaluated trained feedforward and recurrent networks outside of the square training domain. Concretely, we ran the feedforward network with Cartesian coordinates sampled in a $8\pi \times 8\pi$ square, i.e., double the original enclosure wall lengths. For the RNN, we evaluated two cases, one in which the network started in the original square, and was allowed to move outside, and another wherein the network started out anywhere in the enlarged enclosure. In both cases, the RNN was run for a total of 50 timesteps, and responses aggregated over 10000 trajectories.

### 3.7 SUBPOPULATION LOSS EVALUATION

To assess whether different cell types contributed differently to the various loss terms, we evaluated randomly sampled subpopulations of recurrent units on the capacity and distance losses separately. For a given sample, the subpopulation vector consisted of the flattened ratemaps of the selected units. Subsequently, the subpopulation vector was re-normalized to allow for a more direct comparison with the full network loss. As a reference, we also evaluated low grid score units ($n = 29$; grid score cutoff = 0.15) on either loss term. Additionally, we computed a baseline loss by randomly shuffling the bins of unit ratemaps, effectively achieving a spatially random representation. All subpopulations featured the same number of units ($n = 29$, as with the low grid score ensemble), and a total of 1000 samplings were performed.

### 3.8 PERSISTENT HOMOLOGY & LOW DIMENSIONAL PROJECTION

We used persistent homology to investigate whether the learned representation conformed to a simple geometric structure. Specifically, we used the Ripser python library (Tralie et al., 2018) to compute persistence diagrams using spatially flattened ratemaps of unit activity. For the feedforward network, the outermost 20 % of ratemap pixels were excluded to avoid boundary effects, and only units with orientation in the range [0.4, 0.8] were included. This was done to avoid mixing possibly independent modules of units with different orientations. For the RNN ratemaps, the ratemaps of the full domain were used, and every unit was included in the analysis. For the analysis, we set $n\_perm = 500$, and $max\_dim = 2$, with otherwise default parameters.

We further visualized the neural structure of space using UMAP (McInnes et al., 2020) with $n\_neigbhours = 2000$ and $min\_dist = 0.8$ to project ratemaps of the feedforward and recurrent networks down to three dimensions. We colored the resulting 3D point cloud using the first principle component of the ratemaps, similar to Gardner et al. (2022), highlighting important regions. Ratemaps were processed in the same way as for the persistent homology analysis.

### 3.9 FIGURES

Figures were created using BioRender.com.

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

## A    APPENDIX / SUPPLEMENTAL MATERIAL

### A.1    HYPERPARAMETER SEARCH AND LOSS ABLATION

Fig. A1 shows the result of a hyperparameter search in the weighting coefficient $\alpha$, alongside training and evaluation loss for both models, as well as cell types. Note that $\alpha = 0$ and $\alpha = 1$ corresponds to complete ablation of distance and capacity losses, respectively.

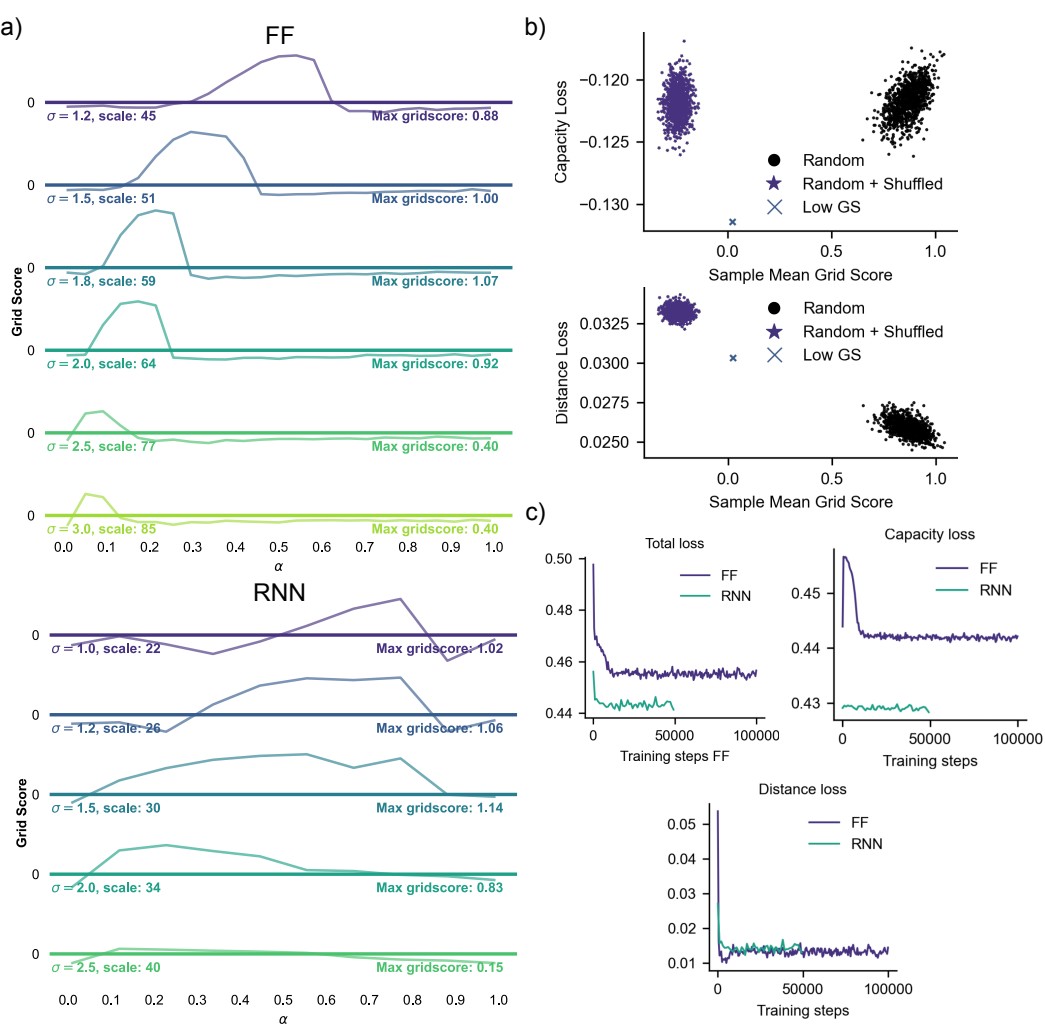

Figure A1: **Hyperparameter search, loss evaluation and training results**. a) Grid score as a function of $\sigma$ and $\alpha$ for FF (top) and RNN models (bottom), where grid scale is calculated for the model with the highest grid score at a particular $\sigma$. b) Loss evaluation on 1000 randomly selected (random), spatially shuffled RNN unit ratemaps (random + shuffled) and low grid score (low GS) subpopulations. c) Loss training history for FF and RNN models.

### A.2    EXTENDED PRUNING RESULTS

Fig. A2 shows ratemaps and error distributions for the recurrent network, in cases where velocity input to all grid and band units are ablated. Notably, path integration error is consistently lower when pruning input to grid than band units, even when every grid unit is velocity-deprived.

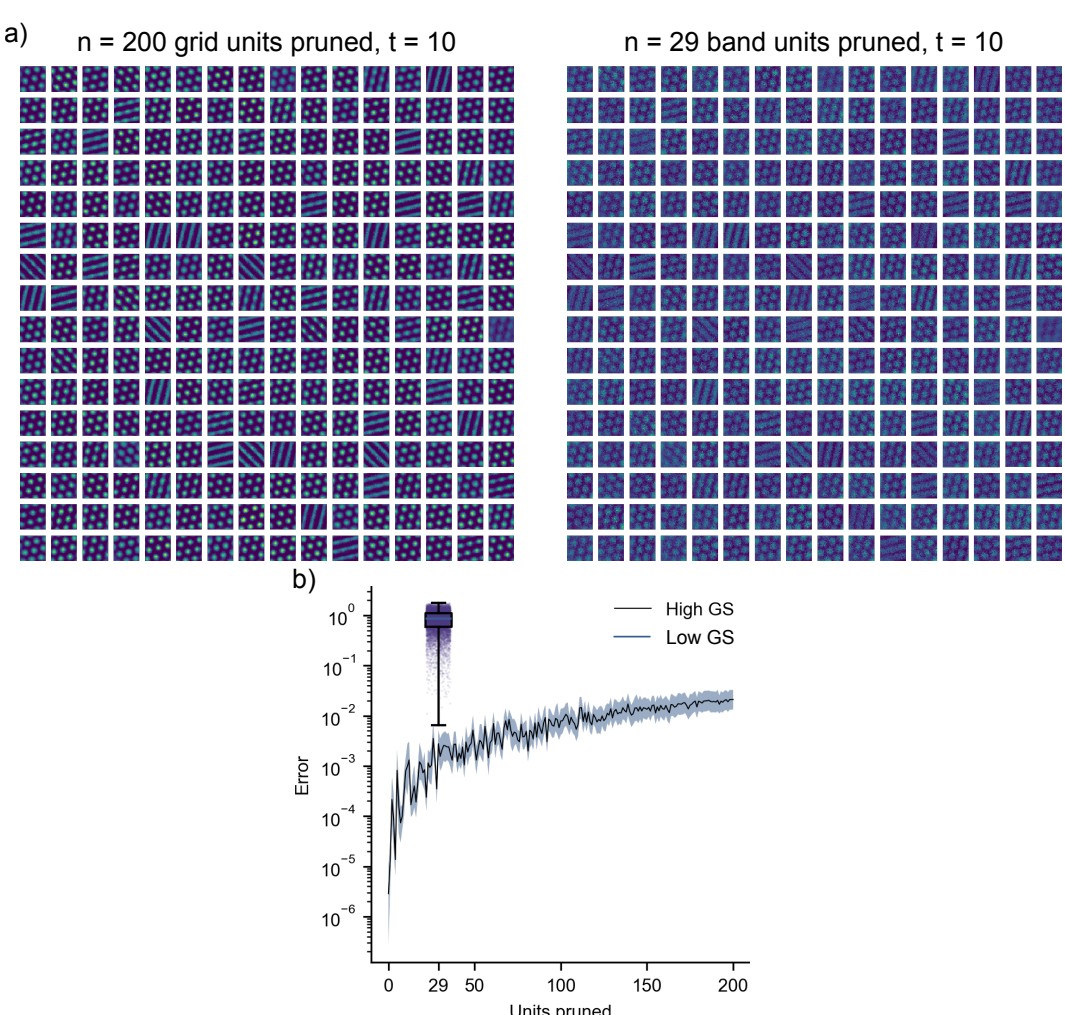

Figure A2: a) Ratemaps for every recurrent unit, after 10 timesteps, when pruning velocity inputs for: 200 grid units (left) and 29 band units (right). b) Final timestep error aggregated across 10000, 10-timestep trajectories as a function of number of pruned high grid score units. Shown is the median (black line) and the 25th and 75 percentile (shaded region. Also inset is a box plot of the corresponding error for pruning $n = 29$ low grid score units, with a corresponding jitter plot of the error distribution. Note the logarithmic scale.

Figure A3: Recurrent unit ratemaps for a network initialized according to (random) uniform distributions.

### A.3 EFFECT OF NETWORK INITIALIZATION

Fig. A3 shows ratemaps of a recurrent network initialized according to a (random) uniform distribution, demonstrating that grid and band-like solutions are still learned without the identity initialization used elsewhere in this work.

### A.4 COMPARISON TO OTHER MODELS

To contextualize our work, we compare it to several recent normative models of grid cells, highlighting both similarities and differences with the works Schaeffer et al. (2023); Sorscher et al. (2022); Dorrell et al. (2022); Dordek et al. (2016); Xu et al. (2022; 2024). Table 1 provides an overview of key characteristics of the models in Schaeffer et al. (2023); Sorscher et al. (2022); Xu et al. (2022), and our approach for both feedforward (FFN) and recurrent (RNN) implementations. It is especially noteworthy in the table that we are the only ones demonstrating emergence of grid-like cells with and without path integration (PI). Additionally, our model achieves high-fidelity grid patterns using a relatively simple architecture, requiring fewer loss components compared to Schaeffer et al. (2023) and omitting place-like encoding/decoding present in Sorscher et al. (2022) and Xu et al. (2022).

Regarding emergence of grid-like cells without path integration, Dorrell et al. (2022) and Dordek et al. (2016) are worth highlighting. Dordek et al. (2016) demonstrates that grid cells can emerge as optimal low-dimensional, non-negative projections (non-negative PCA) of place-cell-like inputs modeled as difference-of-Gaussian radial basis functions. This framework elegantly connects place and grid cells, consistent with their known anatomical connections (Bush et al., 2015). However, this approach does not address normative aspects related to behaviorally relevant tasks, such as path integration or maintaining a distance-preserving representation. In contrast, Dorrell et al. (2022) employs a model constrained to a linear combination of sine and cosine plane waves. While grid-like patterns emerge, the simplicity of this formulation limits the flexibility of the resulting patterns compared to neural network-based approaches. Their functional constraints share similarities with the separation loss in Schaeffer et al. (2023), which encourages distinct neural representations for different spatial positions, but lack explicit distance-preserving objectives present in Xu et al. (2022), Schaeffer et al. (2023), and our model.

Xu et al. (2024) presents a simplified version of their earlier models (Xu et al., 2022; 2023; Gao et al., 2020), removing the reliance on place-cell decoding that is prominent in Sorscher et al. (2022). This refinement aligns more closely with Schaeffer et al. (2023) and our approach by imposing objectives directly on grid representations $\mathbf{g}_t$. However, their initial grid state is not explicitly stated complicating direct comparisons. Similar to Schaeffer et al. (2023) and our work, Xu et al. (2024) incorporates

a hard normalization constraint on grid representations. A key distinction between our FF model and Xu et al. (2024) lies in the ablation of their second assumption—"transformation" (analogous to path integration)—while still achieving grid-like cell emergence. This demonstrates the minimal requirements necessary for producing grid-like cells, emphasizing the connection between grid cells and distance preservation rather than path integration. This is fundamental to our work, as we further show that extending our FF model to an RNN framework, which incorporates path integration, leads to the emergence of band-like representations. We subsequently provide extensive analysis showing that these band-like representations are the primary functional component for path integration.

Whether Xu et al. (2024) also produce band-like cells is not clear. The authors report large ensemble grid scores, suggestive of all-grid representations. However, their models makes use of a heading direction-dependent velocity input projection, which could conceivably produce band-like projections directly in the input. This architectural design is similar to Schaeffer et al. (2023), which also reports high proportions of grid-like units (and some band cells, for certain runs), wherein the recurrent weight matrix is an MLP and a function of the incoming velocity. Given the apparent importance of band cells for path integration in our model, investigating whether these architectures reproduce band-like behaviors could provide valuable insights into how differing model components influence the emergence or suppression of band-like representations, offering a promising direction for future research.

Table 1: Comparison of Grid Cell Models

| Aspect | Schaeffer et al. (2023) | Xu et al. (2022) | Sorscher et al. (2022) | Ours FFN (+ RNN) |
|---|---|---|---|---|
| **Objective** | Sep + Inv + CI + PI | CI + **PI** | **PI** | Distance preservation (+ PI) |
| **RNN Inputs** | Learned RNN weight matrix $W(\vec{v}_t) = \texttt{MLP}(\vec{v}_t)$ from Cartesian velocities | Cartesian velocities $\vec{v}_t$ | Cartesian velocities $\vec{v}_t$ | None (+ Cartesian velocities $\vec{v}_t$) |
| **Initial state** | N/A - the authors report that the network is initialized using a "shared initial state" | Initial position in place cell basis $W_p \vec{p}(\vec{x}_0)$ | Learned linear projection of initial position in place cell basis $W_p \vec{p}(\vec{x}_0)$ | Learned initial representations $\texttt{MLP}(\vec{x}_0)$ |
| **PI architecture** | RNN-like with recurrent weight $W(\vec{v})$ as input: $\texttt{Norm(ReLU(}W(\vec{v}_t)\vec{g}_t))$ | Vanilla RNN and LSTM with ReLU | Vanilla RNN with ReLU | None (+ Normalized vanilla RNN: $\texttt{Norm(ReLU(}W_R\vec{g}_t + W_I\vec{v}_t))$ |
| **Decoder** | None | Linear readout to place cell basis $\hat{\vec{p}} = W_{out}\vec{g}_t$ | Linear readout to place cell basis $\hat{\vec{p}} = W_{out}\vec{g}_t$ | None |
| **Regularization** | L2 Capacity on $\vec{g}_t$ | L2 weight penalty on Linear place cell readout weights $W_{out}$ | L2 weight penalty on recurrent weights $W_R$ | L1 capacity on $\vec{g}$ |
| **Results** | Emerges heterogeneous multi-modular grid-like cells including some place and band-like cells | Emerges high-quality grid-like cells with tunable multi-modularity | Emerges a range of spatially heterogeneous cells, including grid-like cells | Emerges only grid-like cells with tunable multimodularity (+ band-like cells) |

## A.5 SPEED DEPENDENCE OF LEARNED REPRESENTATIONS

As we find that feedforward networks without velocity inputs learn grid-like representations, and that RNNs with velocity input learn additional band-like representations, it is natural to investigate the importance of the velocity signal for pattern formation in the RNN. We therefore trained recurrent networks on 10-timestep trajectory datasets with varying Rayleigh speed scales $s$. The results are presented in Fig. A4, which shows trained RNN unit ratemaps for $s = 0, 0.075$ and $0.75$ (default value $s = 0.15$).

The case $s = 0$ corresponds to no speed, wherein the RNN sits idly at the starting location of each trajectory, not path integrating. Notably, patterns are still grid-like, but there is no apparent band-like tuning. Thus, representations are similar to those of the feedforward network, and again, band structures appear linked to path integration. This also seems to indicate that band responses do not emerge as a consequence of architectural differences between networks, as part of a pattern forming process.

For small speeds ($s = 0.075$), some band-like tuning is again observed. However, for speeds much larger than that used in the default case ($s = 0.75$), a large fraction of units now appear strongly band-like. The pattern, surprisingly, has also turned square, rather than hexagonal, which has been observed in other path integrating models Cueva and Wei (2018). A square pattern may reflect that path integration is more accurately achieved along the cardinal directions of the input velocity vectors, but this observation warrants further investigation.

Together, these results all point towards bands being a necessary component of path integration in recurrent networks, as band-like representations emerge in proportion to the importance of path integration for the task at hand.

## A.6 EXTENDED UNIT RATEMAPS

As noted previously, we find that feedforward networks learn grid-like representations, and that path integrating RNNs learn additional band-like structures. This is shown explicitly in Fig. A5, which shows every unit in two trained models, one feedforward and one recurrent. Notably, all appear grid- or band-like.

When ablating the capacity loss, however, RNN units tend to lose their regular firing patterns. In particular, Fig. A6 shows that low-capacity units have heterogenous firing fields, reminiscent of place fields, in large environments (Park et al. (2011)). While a minority of units do exhibit striped or banded firing fields, none display clear periodic band-like tuning. One might therefore speculate that path integrating band units are inextricably tied to grid cells, and that other cell types, such as place cells, rely on distinct path integration mechanism.

## A.7 DECODING & PATH INTEGRATION PERFORMANCE

Assessing path integration performance in the trained RNN is not straight-forward, due to the highly periodic nature of its learned representations (which causes representations at different locations at distant locations to be ambiguously encoded). However, a network may still be able to path integrate correctly, even without global decodability (as we observe in our model, using the path integration measures in Fig. 2). We therefore develop a simple, one-step linear decoder to uncover this ability, and perform decoding *locally*.

Specifically, we consider a conditional linear decoding of the form

$$\hat{\mathbf{x}}_t = W\mathbf{u}_t,$$

where $\mathbf{u}_t = \mathrm{cat}(\mathbf{g}_t, \hat{\mathbf{x}}_{t-1})$, i.e. a concatenation of the current state $\mathbf{g}_t$ and a previous estimated position $\hat{\mathbf{x}}_{t-1}$, and $W$ is a trainable weight matrix.

We train this decoder in a one-step fashion by feeding in network states and (true) previous-step locations $\mathbf{x}_t$, along trajectories. However during inference, we decode positions iteratively by using the previous position estimate $\hat{\mathbf{x}}_{t-1}$ produced by the decoder, so that the entire decoded trajectory is produced by the decoder. To verify that the decoder is not simply copying the previous location, we compare decodings of 1000-timestep, long-sequence trajectories starting at the origin, to a baseline of always predicting the origin (which would be expected if the decoder simply copied its input).

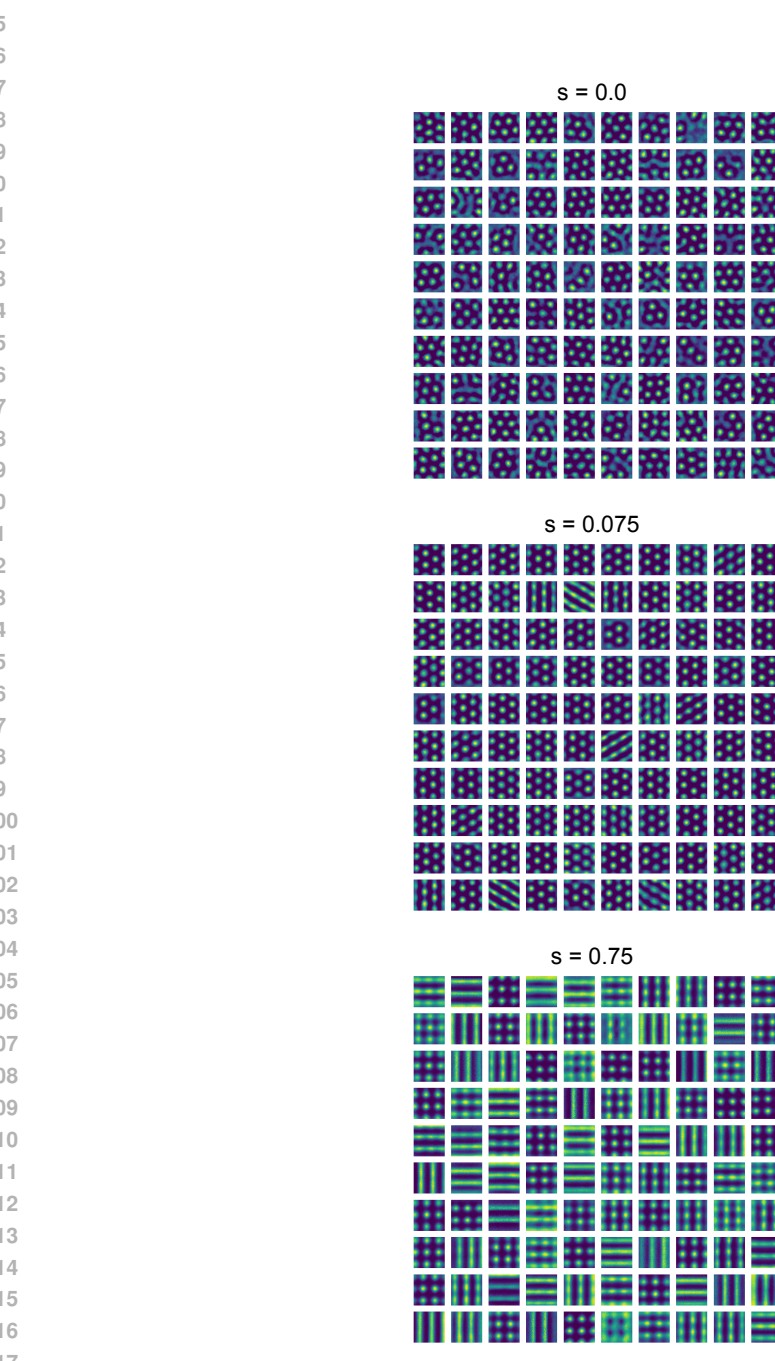

Figure A4: **Speed modulation of learned representations.** Ratemaps of randomly selected RNN units for different Rayleigh speed scales $s$. Higher values of $s$ correspond to faster trajectories, $s = 0$ results in stationary trajectories.

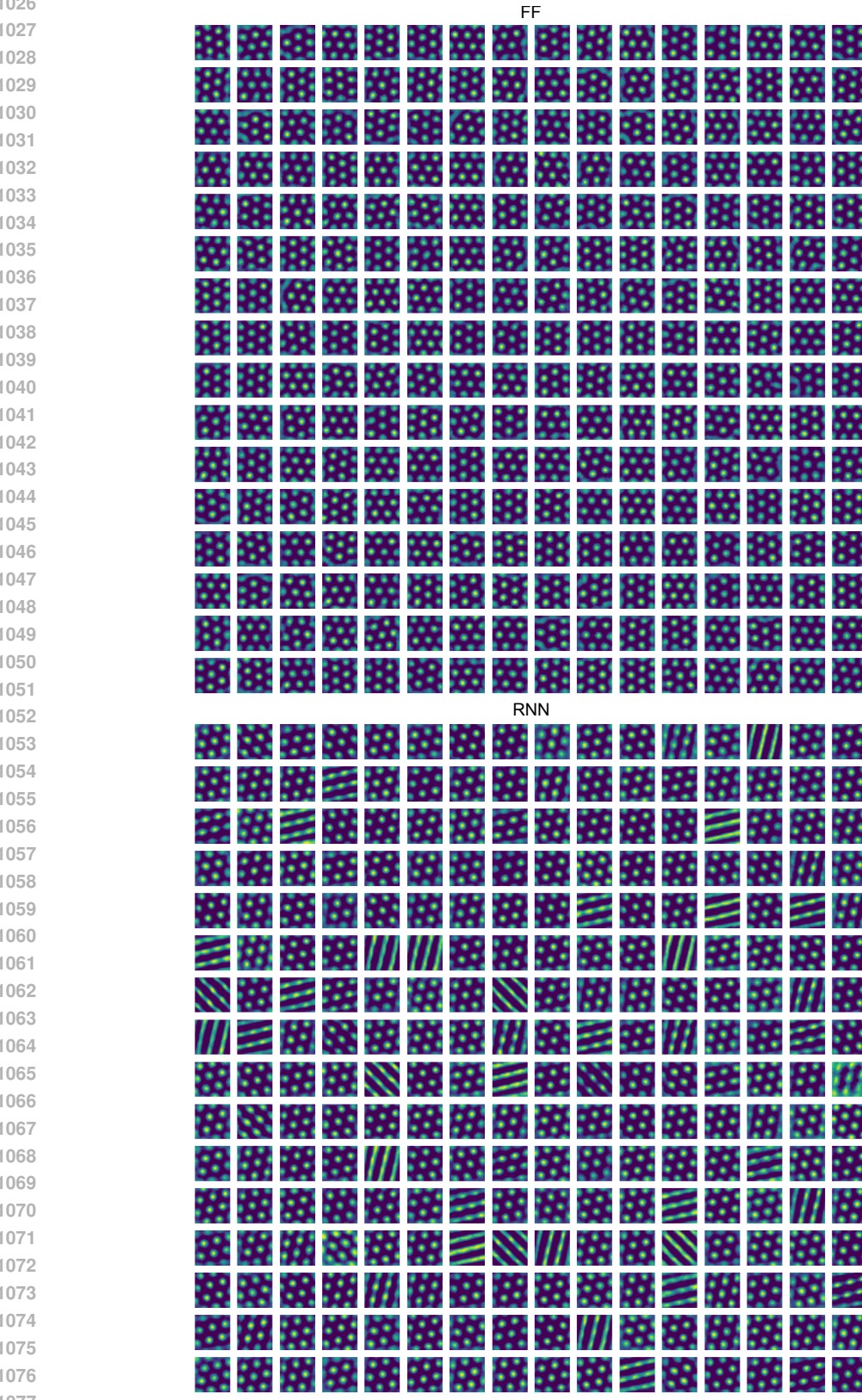

Figure A5: **Ratemaps of all 256 network units.** Ratemaps of every network unit, for feedforward (top), and recurrent (bottom) networks.

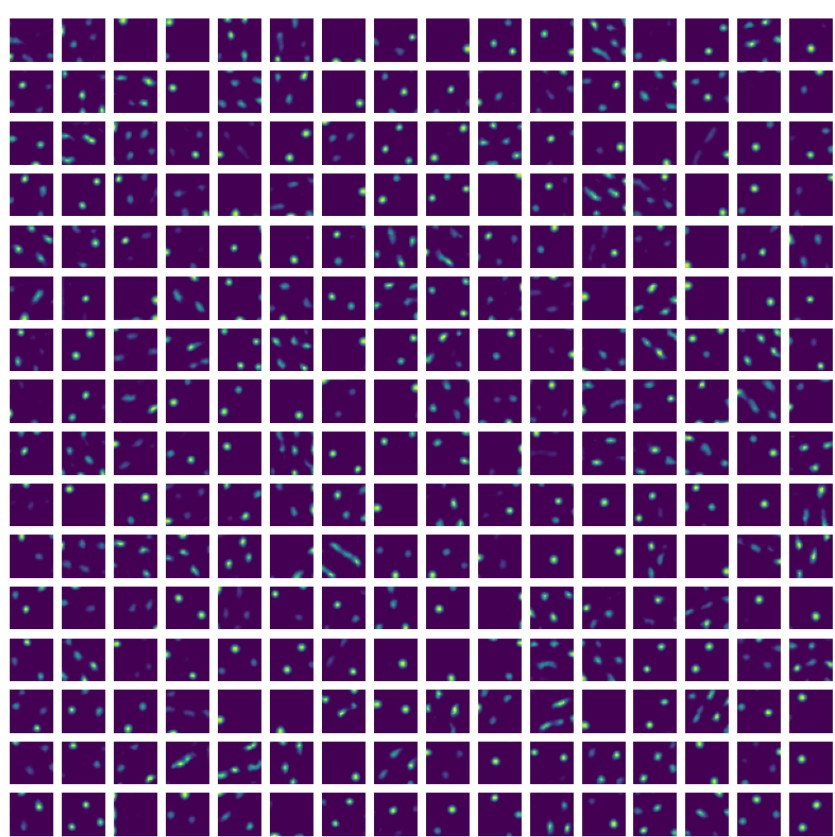

Figure A6: **Low capacity networks learn irregular firing patterns.** All recurrent units, for a network trained with $\alpha = 1$ (no L1 capacity constraint).

The results are shown in Fig. A7a), which demonstrate that not only does the decoder outperform the zero-prediction baseline, but the network is capable of fairly accurate path integration for hundreds of timesteps. Notably, this extends far beyond the training sequence length of 10 timesteps, which is in line with the out-of-box generalization ability observed in Fig. 3c). The training loss in Fig. A7b) also demonstrate that the decoder becomes adept only after thousands of training steps. Fig. A7c) shows an example decoded long-sequence trajectory. Evidently, the decoded trajectory trails the true one for a long time, but shows signs of drift over time.

To explore how path integration ability changes over training time, we trained decoders at varying steps during model training (Fig. A7d)). As expected, path integration ability increases with training time. However, even an untrained network is somewhat decodable (possibly due to the inherent reservoir capacity of larger networks such as ours), but performance degrades considerably faster than trained networks. Another notable feature, is that fully trained networks do *not* appear to be the most capable path integrators, with the 1000-step checkpoint achieving the smallest decoding error.

Inspired by this, and our findings that band units are most important for path integration in trained networks, we examined network representations in terms of their *band score*. Following Redman et al. (2024), the band score is given by

$$b(G) = \max_{k_x, k_y} \text{Corrcoeff}(G, S(k_x, k_y)),$$

where $k_x, k_y \in \{0, 0.1, 0.2, ..., 2\pi\}$ are spatial frequencies in the $x$ and $y$ directions, corrcoeff the Pearson correlation coefficient, and for $G$ we use the autocorrelogram of the ratemap of a particular network unit (to ensure $G$ is centered), and $S(k_x, k_y)$ a 2D sinusoid whose frequency is given by $k_x$ and $k_y$.

Intriguingly, we find that band scores indeed correlate with path integration ability (compare Fig. A7d) and e)). In particular, ensembles with greater band scores tend to perform better; compare e.g. the fully trained model, and the network trained for 1000 steps. As can be seen from inset ratemaps, ratemaps at this particular step are more square, and highly band-like.

As a final remark, this correspondence should not be taken to mean that path integration is necessarily performed by band units; it could for instance reflect that such units are more linearly decodable, for instance. However, these findings resonate with our other results (which do not explicitly rely on a linear decoder), again hinting at a connection between path integration ability and band-like representations.

## A.8 AN L2 CAPACITY CONSTRAINT INDUCES HETEROGENEOUS REPRESENTATIONS

In this work, we have shown that distance preservation and an L1 capacity constraint is sufficient to induce grid patterns in feedforward and recurrent networks. However, this choice was in part motivated by other work (Schaeffer et al., 2023), that utilizes a similar L2 constraint. To demonstrate why an L1 constraint, rather than an L2 appears more conducive to grid-like representations, we trained multiple networks with varying L2 capacity constraints.

The resulting ratemaps are shown for both FF and RNN networks in Fig. A8. Notably, recurrent unit responses do appear hexagonal and grid-like (as well as band-like) for appropriate $\alpha$ values. However, representations are more irregular than what we observe for an L1 constraint, with some units being mainly silent, some exhibit spurious firing fields, and others incomplete grid firing fields. Comparing to results for L1 capacity, grid cells are known for their tendency to fire all over the recording environment (in symmetric geometries), and exhibit persistent activity across different environments, suggesting an L1 constraint may be more appropriate.

Unlike for the L1 case, we could not find hyperparameter values $\alpha$ that resulted in grid-like representations in the feedforward network using an L2 capacity constraint. Rather, the learned representations tended to be sparse. Thus, an L1 capacity appears to favor distributed representations, aligning well with grid cell properties and potentially contributing to more robust representations. However, it should be pointed out that the presence of path integration induced band- and grid-like representations, also for the L2 constraint; suggesting that it may enhance pattern formation, and that band and grid patterns are better suited for simultaneous path integration and distance preservation than heterogeneous ones.

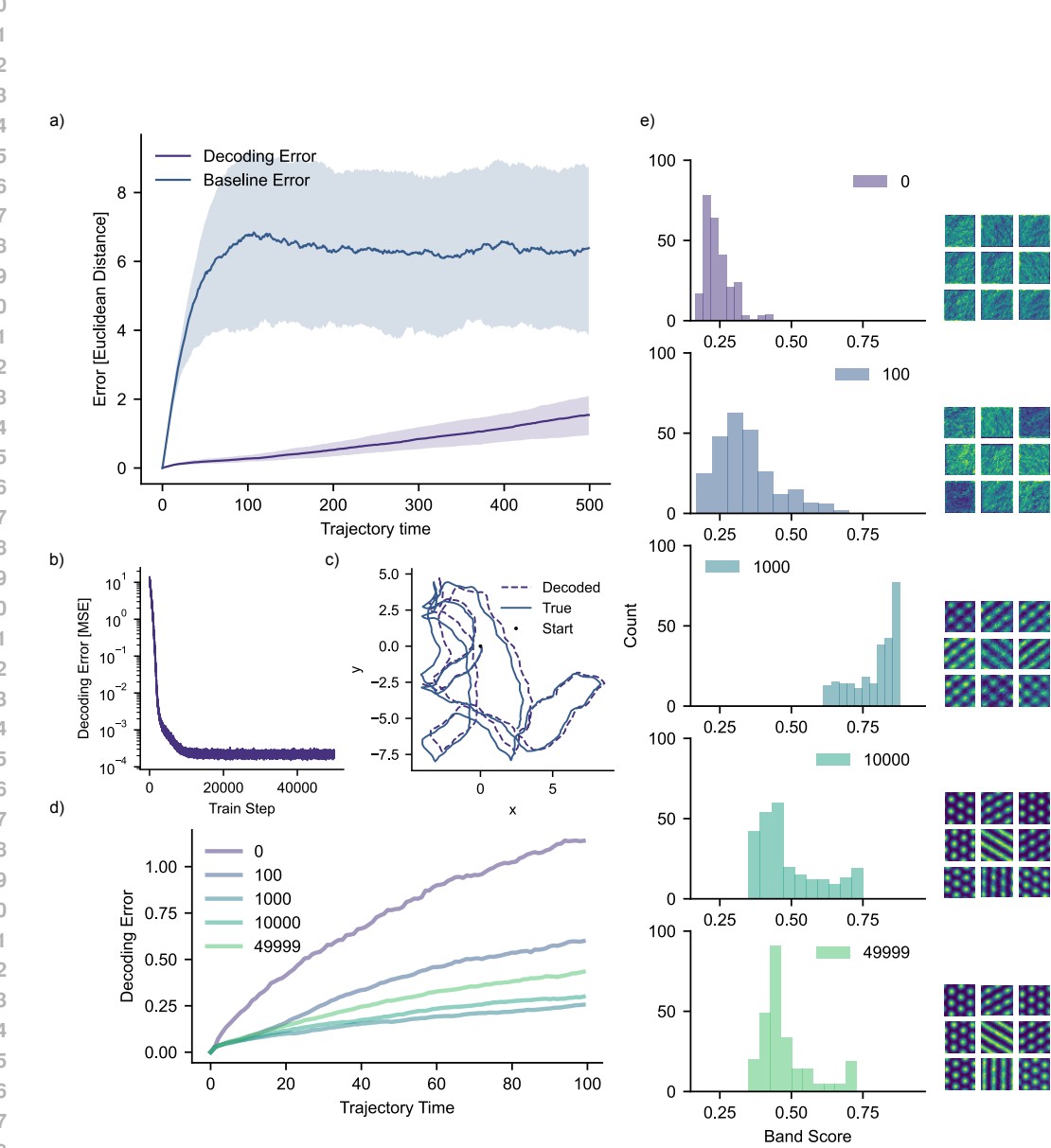

Figure A7: **Recurrent network path integration performance**. a) Path integration error (Euclidean distance between true and decoded trajectories) for the one-step linear decoder. Shown is the median error, for a trained RNN model, and a baseline, all-zero prediction, for 1000 trajectories all starting at the origin. Shaded regions indicate the inter-quartile range. b) Training loss for the one-step decoder. c) An example decoded true trajectories, alongside the corresponding true trajectory. d) Decoding error over trajectory time, as a function of model training length (legend indicates training step; 49999 denotes a fully trained RNN). e) Ensemble band scores corresponding to the training steps in d). Also inset are randomly selected unit ratemaps at the indicated training step.

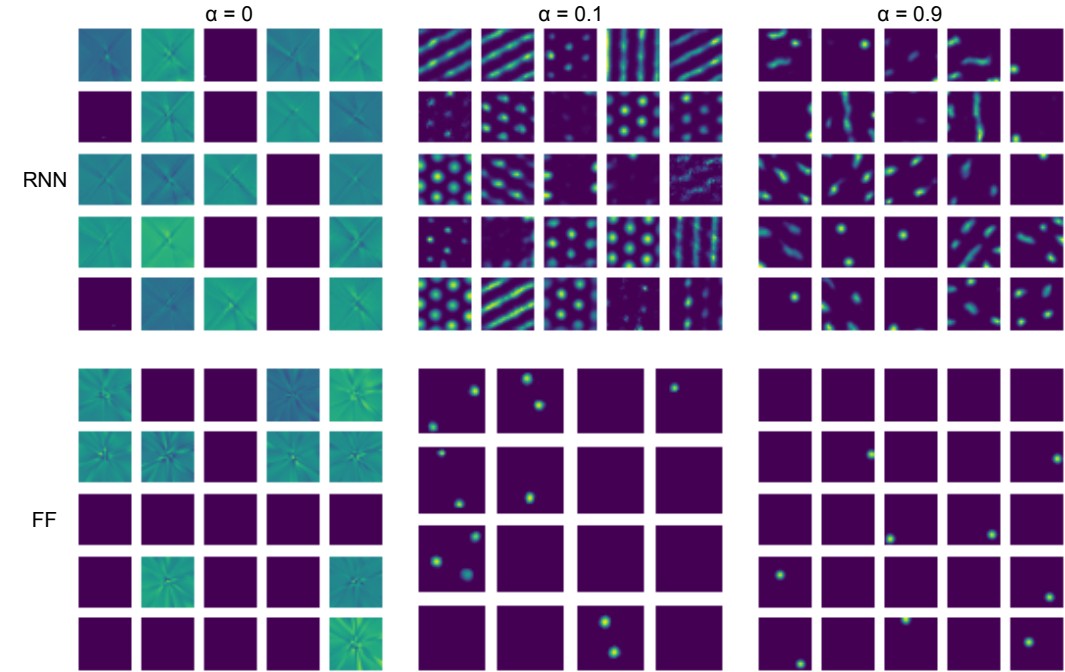

Figure A8: **Effects of L2 capacity constraints on learned representations.** Ratemaps of randomly selected units when the L1 capacity constraint is exchanged for an L2 constraint. Shown are responses for RNN (top) and FF (bottom) network units for varying loss weightings. $\alpha = 0$ corresponds a pure capacity loss, $\alpha = 1$ to a pure distance preservation loss.

## A.9 GRID SPACING AND FIRING FIELD SIZE

The grid search in A1 unveiled that the envelope scale parameter $\sigma$ dictates the spacing of the grid pattern. This is shown explicitly in Fig. A9, wherein feedforward grid fields are shown to become more distantly spaced with increasing $\sigma$.

A related, interesting quantity is the grid firing field size. How this quantity is determined in our model, is not entirely obvious. However, because of the close correspondance between neural distances and physical ones, we note that we can modulate the scale of the pattern, by introducing a factor $\rho$ into the loss function,

$$\mathcal{L} = \alpha \mathbb{E}_{t,t'} \left[ e^{-\frac{1}{2\sigma^2} \|\mathbf{x}_t - \mathbf{x}_{t'}\|^2} \left( \rho \|\mathbf{x}_t - \mathbf{x}_{t'}\| - \|\mathbf{g}_t - \mathbf{g}_{t'}\| \right)^2 \right] + (1-\alpha) \mathbb{E}[l_{cap}(\mathbf{g}_t)],$$

so that the representation learns to represent distances scaled by a factor of $\rho$. This is similar to the conformal scaling factor used by (Xu et al., 2022).

Ratemaps of feedforward units trained to minimize this slightly modified loss are also shown in Fig. A9, which demonstrates how firing fields become larger with decreasing $\rho$. Following (Xu et al., 2022) and (Xu et al., 2024), this control over learned representations allows us to directly introduce multiple modules in an interpretable way, by partitioning the network into distinct modules, or allowing $\rho$ to be a unit-specific trainable parameter.

Thus, grid field size and spacing can be readily understood in our model: field sizes reflect the ratio between neural and physical distances, while grid spacing reflects the scale at which distances should be accurately represented (for a fixed field size).

This result also highlights an important, but subtle difference between distance preservation, and a conformal isometry requirement: While both enable faithful distance computations by integrating the (flat) metric along neural trajectories, demanding distance preservation allows distances to be computed directly by comparing two population vectors (while in the range where this is valid),

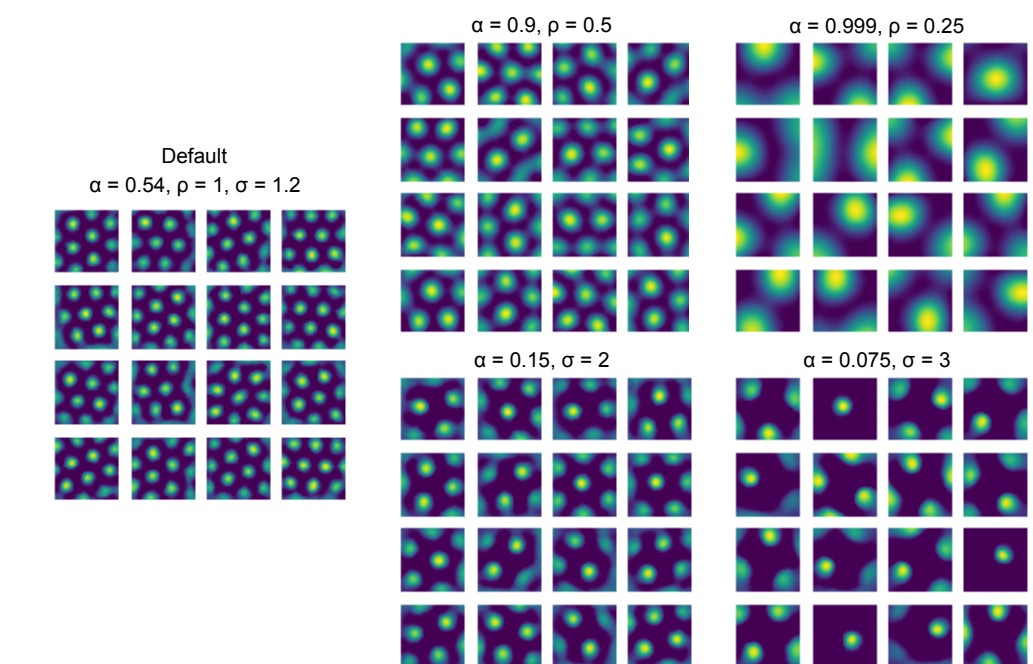

Figure A9: **Grid spacing and firing field size can vary independently.** Ratemaps of randomly selected units for different parameter combinations in $\sigma$ and $\rho$. Shown is the default FF case (left), decreasing $\rho$ (rightmost, top row), and increasing $\sigma$ (rightmost, bottom row).

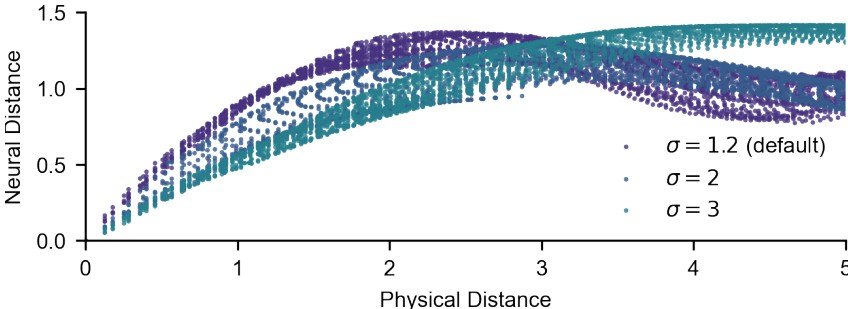

Figure A10: **Euclidean distances in representation vs. space.** Distances between population vectors of feedforward networks trained with varying scale parameters, $\sigma$, versus distances in the arena. States and locations are sampled from a square grid, i.e. a ratemap.

which could greatly simplify distance computations. However, it should be noted that this is also true to first order for a conformal isometry.

That our model preserves Euclidean distances (and preservation is determined by $\sigma$), is showcased in Fig. A10, where representational and physical Euclidean distances are compared directly. As shown, a larger value of $\sigma$ induces a right-shift in the distance plot, and the relationship between the two is near-linear for longer, indicating that distances are preserved.

Exploring the scale at which distances are preserved in biological grid cell data, could make for an interesting comparison between our model and others, and could possibly account for variations in grid spacing, which we find coincides with the scale at which (Euclidean) distances are preserved in the representation.

