# OpenReview forum: "Self-Supervised Grid Cells Without Path Integration"
_ICLR.cc/2025/Conference — Submitted to ICLR 2025_

### Official Review · Reviewer_dB3b · 2024-10-31

**Soundness:** 3
**Presentation:** 3
**Contribution:** 2
**Rating:** 6
**Confidence:** 4

**Summary:**

This paper proposes a self-supervised objective function demanding distance preservation subject to a capacity constraint, as a model of grid cells in the MEC. The authors used the objective to train two different neural network architectures and found that grid cells emerged in both models. The authors have also performed a pruning experiment to understand the role of grid cells and band cells in path integration and reached the conclusion that band cells instead of grid cells are responsible for successful path integration. They also performed a thorough investigation into the pattern formation and generalization capabilities of the models.

**Strengths:**

I enjoyed reading the experiments of this paper, which are very comprehensive. The motivation and details of each experiment are also clearly presented. I particularly like the authors’ discussions on the role of band cells as this cell type has been understudied in many of the previous works on modeling grid cells. The Methods section is also very clear and well-presented.

**Weaknesses:**

- Writing: Despite the clear presentation of each experiment, I believe the paper can be improved by adding subsection titles to each of its 3 experiments. From row 107 to 154 the paper is clearly presenting the result that grid cells do emerge in models trained with the proposed objective, which can be put under one subtitle. Similarly, from row 155 to row 199 the paper is talking about pruning and from 199 onwards it is presenting results on pattern formation, connectivity and generalization of the models. It would be nice to have subtitles for readers to navigate through the experiments.
- Comparison to Schaeffer et al. 2023: A critical component that is missing from this paper is a formal comparison to other self-supervised learning models for grid cells, in particular Schaeffer et al. 2023, as it bears a lot of similarities to the self-supervised loss proposed in this paper. From what I understand (also what the authors claimed), the loss in this paper is simplified compared to Schaeffer et al. 2023 as it only keeps the distance preservation and capacity terms, but discarded the path invariance and conformal isometry terms. I have the following questions (the authors don’t have to run extra experiments to answer these questions; some conceptual clarification in the introduction should suffice. Although experiments are welcomed if they can provide further clarification):
    - What is the motivation for removing these two terms except for simplicity?
    - Why does it still work without these terms (I remember Schaeffer et al. performed an ablation study and found that path invariance is critical to the emergence of grid cells)?
    - Is it related to the L1 capacity constraint? Does it introduce path invariance implicitly?
- Novelty (related to previous point): If I understand correctly, the novelty of this paper is 1) a new self-supervised model for grid cells and 2) using the model to understand the necessity and sufficiency between grid cells and path integration. The second novelty is a valid and interesting one and I like it a lot; however, without a formal comparison to Schaeffer et al. 2023, I found it hard to understand why the authors used their new model instead of the other models (Shaeffer et al. 2023, or even path-integrating RNN models). This is particularly important given that the authors claimed in the Summary “Our approach, featuring a minimal model, has allowed us to isolate key factors contributing to the emergence of grid-like representations” - how is the isolation of key factors unique to the proposed model?

**Questions:**

- I wonder if the significance of band cells to path integration has something to do with the speed of the simulated agent? In my own experiments with path integrating RNNs, I found that band cells are more likely to emerge with high-speed agents. Intuitively this makes sense because if an agent moves too fast then two firing peaks of a grid cell might be linked together. Feel free to do some experiments with speed if you have time/space in the paper!
- Fig2d: This figure is quite hard to understand. I understand what each column means (time step), but what does each row mean? Do they correspond to c, e, and f? Maybe I missed it but I didn’t find any information about the meaning of each row in the main text.
- Fig3c: the caption for this panel has some typo.

Overall, I'm giving a score 5 to this paper. However, if the authors can clarify the difference to Schaeffer et al., 2023 and highlight the key factors in their loss that 1) enabled the emergence of grid cells without path invariance and conformal isometry and 2) enabled them to perform the pruning experiments that explored the role of grid/band cells in path integration, I would be happy to raise the score to 6.

---

> ### Author Response · Authors · 2024-11-23
> **Reply**
>
> On a general note: thank you for your kind feedback.
>
> On presentation: Thank you for raising this point: We agree, and  will add subtitles to each result.
>
> On comparisons with other work: As we write in response to other reviewers, we will add a discussion on how our work relates to other models, and the role of L1 and L2 capacity constraints.
> The main motivation for utilizing these two terms, we argue, is that (similar to Dorrell et al., and Xu et al.) spatial representations should accurately reflect distances in a neighborhood around your current location, and that representations should be efficient (using as little of the state space as possible, similar to Schaeffer et al.), and distributed, which also affords robustness. Otherwise, we would highlight simplicity as one of our main modeling motivations, as a huge range of models with multitudes of different regularization strategies, constraints and architectures all report grid-like representations.
>
> Regarding path invariance: This might be implicitly enforced by (non-infinitesimal) distance preservation in our model:
> If an RNN approaches a location from different directions, then the distance between representations should only depend on spatial distance (not incoming direction). However, this could also be the case for the CI loss formulation in Schaeffer et al., which could point to the path invariance-term being somewhat redundant, at least in that model. However, it could be that path invariance is required for learning consistent representations in that model, due to e.g. long sequences (whereas CI is only required for short distances).
>
> Due to the many interacting loss terms, we are unsure about the exact contribution of the L2 capacity constraint in Schaeffer et al.; but they suggest it is linked to multimodularity, and spatial tuning.  They do report that ablating separation loss causes loss of spatial tuning, so it might be that their scale separation term forces similar representations to be aligned (maximizing differences between distinct states), inducing distributed representations, which we find can be introduced directly by an L1 constraint. We will, however, discuss this more in the revised paper.
>
> Concerning the choice of model: There are several reasons why we employ our model, instead of others. The first, as mentioned, is simplicity: comparing to e.g. Schaeffer, our model features just two loss terms, a vanilla RNN architecture (compared to an RNN where the weights are a velocity-dependent MLP), and we observe that the RNN only features one module of grid cells. The second is that the same objective is directly “portable” to non-path integrating networks (i.e. a feedforward network), which are key to demonstrating that path integration is not necessary for grid emergence. It is unclear how that could be done using Schaeffer et al.’s model, which requires path integration e.g. to define path invariance.
>
> On speed: This is a great suggestion, and we will add examples of models trained with varying trajectory simulation speeds. In particular (as was also asked by another reviewer), we are interested in what happens when the speed goes to zero, and we will add higher-speed agents. Notably, zero-velocity RNNs do not exhibit bands, similar to the feedforward network, potentially ruling out architectural differences as the cause of these patterns.
>
> On Fig. 2d: thanks for pointing this out, we will improve upon the caption, and the figure legibility. It shows the path integration error at every timestep, for subsamplings (of size n = 29 units) of the RNN population. It demonstrates that there is a linear relationship between error and (subpopulation) grid score at every timestep, showing that grid cells (high grid score units) are consistently less important than band cells, and apparently that errors compound when pruning low grid score units, indicating that path integration has been disrupted.
>
>  On Fig 3c: Thanks for catching this. We will correct it.

---

> ### Author Response · Authors · 2024-11-27
> **Revision**
>
> We have added several (supplementary) analyses to address your concerns to the revised text, including
> - subtitles for clarity
> - A longer comparison between our model and others (Appendix A4)
> - An analysis of the speed dependence of learned representations, showing that RNNs trained with zero-speed trajectories do not learn bands. Also, for larger speeds, band cells become more prominent, and patterns more square. (Appendix A5)
> - Added analyses that showcase the effects of model parameters (Appendix A9, A6, A8)
> - Added a linear decoding scheme as an alternative analysis of path integration ability, and showcasing that path integration ability correlates with band tuning in the representation during training. (Appendix A7)
> - improved captions and removed existing typos.
> - Highlighted a potential difference between a CI requirement and our distance preservation (Appendix A9)
>
> Together, we hope that these improvements meet your concerns. Please reach out if you have more questions.

---

> > ### Comment · Reviewer_dB3b · 2024-12-01
> >
> > I appreciate the amount of additional experiments conducted by the authors in such a short period of time, and the clarification on comparison to other models. The discussion on L1 vs L2 is much clearer now, although I do think the paper could do better with this (important) discussion placed in main text rather than appendix. However, I understand the time for revision is (very) short given the many additional experiments and paper space is limited, so I'm happy about the current version and raised the score to 6. I hope the authors can think about placing the comparison with other models (not the full comparison table, but a few sentence summarizing the table) and especially that between L1 and L2 to the main text in their camera-ready version or any further submissions.

---

> > > ### Author Response · Authors · 2024-12-02
> > >
> > > Thank you for your positive feedback and for raising your score. We’re very happy that you appreciate the additional experiments and clarifications.
> > >
> > > We agree that including the discussion on L1 versus L2 regularization and summarizing the comparisons with other models in the main text would enhance the paper. While page limitations make this somewhat challenging in the current version, we will rearrange so that we can include these points in the main text for the revised submission.
> > >
> > > Thanks again.

---

### Official Review · Reviewer_KxAv · 2024-11-02

**Soundness:** 2
**Presentation:** 3
**Contribution:** 1
**Rating:** 3
**Confidence:** 4

**Summary:**

This paper explores the functionality of grid cells and investigates why they emerge in biological systems using an RNN-based deep learning model with a specially designed loss function. The loss function consists of two components: the first enforces the Euclidean distance between RNN representations to reflect corresponding spatial distances within a defined range, effectively aiming for a locally isometric mapping between physical locations and neural representations. The second component is an L1 norm on all neural activations, intended to reduce overall neural energy consumption. By training the RNN with this loss, the authors observe the emergence of grid cells and band cells similar to those observed in experiments. Two training paradigms are used: one with velocity input and one without. The findings suggest that the emergence of grid cells is independent of path integration. Additionally, selective removal of velocity input from neurons with higher grid scores showed that path integration errors remained unaffected, suggesting that grid cells might not be directly involved in path integration tasks.

**Strengths:**

The loss function design is novel in its approach to generating grid cells, explicitly incorporating spatial distance representations within the loss. This addition reinforces existing theories positing grid cells as spatial metric providers.

**Weaknesses:**

1. **Novelty:** While the specific mathematical form of this loss function is novel, the core idea has been implemented in slightly different forms in previous studies. For example, in Dorrell (2022), the loss function encourages distant physical points to have more different neural representations. Although the formulation differs, the underlying concept is similar. Additionally, the observed phenomena regarding grid cells and path integration, such as grid cell emergence independent of path integration tasks and band cells being more likely to perform path integration than grid cells in RNNs, have been previously reported (see Sorscher et al. 2022, Schøyen et al., 2023).

2. **Grid Cell Characteristics:** The grid cells emerging in the RNN exhibit a single spacing and varying orientations, which contradicts experimental observations. In biological systems, grid cells are organized into modules with similar orientations and distinct spacings.

3. **Loss Function Design:** The design of the loss function itself raises concerns. If grid cells are intended to maintain an isometric mapping, why is this restricted only to the local environment? This constraint appears to have been added to explain the hexagonal firing patterns of grid cells, yet this assumption lacks biological justification and complicates the functional interpretation of grid cells. This restriction might also explain why the model produces grid cells with a single spacing, further detracting from its biological plausibility.

**Questions:**

In Figure 2b, it seems that only two neuron types emerge in the model: grid cells and band cells, with classification based on grid score. However, in other studies, such as Sorscher et al. (2022), many other neuron types have been observed, including neurons without clear spatial periodicity. I am curious whether such neuron types are indeed absent from the model proposed in this paper. If so, could the authors discuss why this might be the case? A comparison between model designs to highlight factors driving these differences could be interesting. Alternatively, if these neurons are present but not mentioned, I recommend that the authors clarify this in the manuscript.

---

> ### Author Response · Authors · 2024-11-23
> **Reply**
>
> We thank the reviewer for their perspective. While our strategy for arriving at grid-like representations is not new, and builds upon Dorrell et al., Xu. et al. (2022) and Schaeffer et al. (2023) (which we reference when introducing the model), we argue that our main modeling novelty, is producing a highly simplified model of single grid module emergence, using only two distinct loss terms. Our main novelty, however, is applying such self-supervised methods to tackling the problem of whether grid cells do path integration. In comparison, Sorscher et al. and Schøyen et al. utilize a model tasked with predicting place cell activity that exhibits a range of emergent cell types. In our case, our model does not require any place cell labels, and only exhibits two distinct cell types: Grid cells, and, if path integration is present, additional band cells. It is this separation into only two cell types that allows us to perform functional assignments, even in the self-supervised setting. We would also like to highlight our modeling and pruning approach as an important novelty in two ways. First, we prune velocity projections to the recurrent units compared to Schøyen et al. and Nayebi et al who ablate recurrent units. This pruning strategy has the advantage of not confounding recurrent homeostasis effects with path integration performance. Second, we evaluate path integration on the recurrent representation as it mitigates the possibility that path integration disruption is confounded by disruption of the readout into place-like labels present in the model of Sorscher et al. We will be happy to add this as part of the discussion of our paper.
>
> Regarding that the model only exhibits a single module: We agree that biological grid cells arrange in multiple modules with differing orientations and scales. However, we want our model to be highly simplified, precisely to uncover questions of path integration within modules. As we show in Fig. A1, however, the scale of the learned representation depends on the scale parameter sigma. Thus, multi-modular grid cells can be introduced by partitioning the network into different scales, similar to Xu et al. (2022). If less control of the learned representations is desired, more strongly self-supervised objectives is likely required, akin to Schaffer et al. (2023).
>
> On the choice of loss function: We thank the reviewer for raising this concern: The key property is that distances should be preserved in a neighborhood around your current location, a point which was also raised by Dorrell et al. (2022). Regarding why isometric mapping is restricted to the local environment, although this property is evaluated locally, its required everywhere in the environment (expectation over spatial positions in Eq. 1), making it a global property. Hence distance preservation is preserved everywhere, e.g. (global) distances computed through integration. And we agree that this is likely what induces unimodular grid patterns - because we dictate the scale at which distances should be preserved. As such, it could be interesting to demand e.g. scale-invariance/equivariance, or directly introduce multiple modules in our modules by partitioning the network into discrete scales. As previously described, however, we think that unimodularity is useful in our model, as it makes the resulting network simple, and easy to probe for path integration.
>
> On comparison with other models; thank you for raising this point; as also requested by other reviewers,  we will add a discussion comparing our model to other recent normative models. And while we do reference Sorscher et al., we do not consider a thorough comparison between our model and Sorscher’s as relevant as comparing to the more recent and similar models of  Xu et al. and Schaeffer et al, upon which our work is chiefly based. But key differences include (i) imposing path integration directly on the recurrent representations (similar to Schaeffer et al. 2023) compared to Sorscher’s place-cell readout (which has been critiqued in e.g. Schaeffer et al. 2022) and RNN-initialisation mechanism, (ii) our model features unit-normalization (iii), our model uses a L1 capacity constraint rather than L2 weight decay (iv).
> We will also include ratemaps of all units present in our models in the appendix - showing that our model cell types is limited to grid-like cells and band cells (for the RNN model), we realize that this did not come across clearly before - thanks again.

---

> ### Author Response · Authors · 2024-11-27
> **Revision**
>
> Besides the points mentioned above, we have included several additional analyses in the revised manuscript (as appendices), including
>  - A comparison of our model and others (Appendix A4)
>  - Ratemaps of all units in both networks, demonstrating that all units are either band- or grid-like.  (Appendix A6)
> - Higlight a distinction between our distance preservation and a CI requirement (Appendix A9)
>
> We hope this answers your questions, please reach out if you have other concerns.

---

> > ### Comment · Reviewer_KxAv · 2024-11-30
> >
> > Thank you for your detailed response and for addressing my concerns. After reviewing your reply, I still have significant reservations about some of the key points, as I believe they have not fully addressed my critiques.
> >
> > 1. Biological Plausibility of the Model:
> >
> > If the primary goal of your work is to investigate whether grid cells perform path integration in biological systems, the types of neurons emerging from your RNN model should at least approximate the diversity of neural types observed in the medial entorhinal cortex (MEC). However, as you acknowledged, your model only produces grid cells and band cells, with no other neuron types present. While you argue that this simplification is an advantage, as it allows for functional assignments, it also significantly diminishes the biological plausibility of the model. Biological MEC includes various neuron types, many of which are not periodic in their spatial tuning. The absence of such diversity in your model makes it difficult to draw meaningful conclusions about the functional roles of grid cells in real biological systems.
> >
> > Furthermore, the single-module nature of the grid cells in your model does not align with the modular structure observed in biological systems, where grid cells form modules with distinct spacings but consistent orientations. This discrepancy weakens the claim that the findings from your model, such as the potential role of grid cells in path integration, can be extrapolated to biological systems. In this context, the simplicity of your model, while methodologically convenient, undermines its ability to support hypotheses about real-world neural mechanisms.
> >
> > 2. Pruning Velocity Projections:
> >
> > You highlight the novelty of your pruning strategy, which removes velocity projections to recurrent units, and contrast it with other studies that ablate recurrent units. However, I do not find this aspect particularly novel or compelling. As you mention, Schøyen et al. (2023) already explored the impact of velocity pruning on grid cell function in path integration tasks. While your implementation may differ slightly, the conceptual approach is not unique, and the lack of other neuron types in your model further limits the broader implications of your findings.
> >
> > In conclusion, while I appreciate the clarifications you provided, I still find the biological relevance of your model's results to be limited. The simplifications introduced in your model make it difficult to convincingly argue that the observed behaviors of grid cells in your RNN are indicative of their roles in real biological systems.
> >
> > Thank you again for your efforts, and I hope my feedback is helpful in further refining your study.

---

> > > ### Author Response · Authors · 2024-12-02
> > >
> > > Thank you for your thoughtful and detailed feedback. We appreciate the opportunity to address your concerns regarding the biological plausibility of our model and the novelty of our pruning strategy.
> > >
> > > We acknowledge that our model simplifies the complex neural landscape of the medial entorhinal cortex (MEC) by not including other simulated neuron types, such as head direction cells, border cells, or object vector cells, which are known to contribute to spatial navigation in biological systems. Additionally, our model is rate-based and uses backpropagation for training, which raises questions about its direct transferability to biological systems. However, the purpose of our model is not to construct a comprehensive model of the MEC, but to investigate the types of representations necessary for encoding distances, with or without path integration.
> > > By adopting this minimalistic approach, we aim to isolate and better understand the fundamental mechanisms underlying specific aspects of spatial navigation. We find that only grid and band cells naturally emerge within this framework, suggesting that these cell types are central, and sufficient to solve these computational tasks. If other neuron types were required for this task, we believe they would emerge in the model as well.
> > >
> > > Simplifying the network also provides the advantage of reducing confounding factors, allowing us to focus on core computational principles. While we agree that incorporating the full diversity of MEC neuron types would improve biological realism, we do necessarily believe it would provide additional insights into the computational roles of grid and band cells in path integration and distance representation. Again, if this were the case, we believe that other required cell types would emerge in response to the task, similar to band and grid cells in our model.
> > >
> > > Of course, grid cells could serve other purposes in conjunction with different cell types, which is a fascinating topic for future research but beyond the scope of this work. However, minimal models like ours serve as tools for breaking down complex systems into their essential functional components, which can be used to generate hypotheses that may guide future empirical studies. This approach aligns with the broader perspective in theoretical and computational neuroscience that emphasizes the utility of simplified models for foundational insights.
> > >
> > > Regarding the absence of multiple modules, we demonstrate in our new analyses that we can readily modulate the firing field size of the grid pattern. This capability allows us to partition the RNN into multiple modules (either directly or by training), as has been shown in other work (e.g., Xu et al., 2022/2024). However, we do not believe that introducing multiple modules would change our fundamental results; rather, we expect to observe the co-emergence of bands at corresponding scales.
> > >
> > > On the topic of our pruning strategy, we appreciate your reference to Schøyen et al. (2023). While their work also explores pruning and its impact on grid cell function in path integration tasks, our approach is distinct in several key respects. Specifically, our strategy targets the velocity projections to recurrent units, minimizing homeostatic confounding, which can complicate the interpretation of network path integration ability following pruning. In contrast, Schøyen et al. prune network units directly, which makes it difficult to disentangle the disruption of path integration from broader off-target effects that could destabilize the network.
> > > Furthermore, our model reveals a clear functional differentiation between grid cells and band cells. We observe that band cells emerge only when the network is trained on path integration tasks, whereas grid cells emerge when to preserve distances, with or without path integration. This suggests that band cells may play a more critical role in path integration than grid cells, a conclusion that complements and extends the findings of Schøyen et al. (2023). By showing that our model naturally develops these distinct cell types in response to a self-supervised training objective, we provide additional evidence for their specialized roles in spatial computation.
> > >
> > > We are committed to revising our manuscript to better clarify the purpose, limitations, and implications of our study and to address the concerns you have raised. Thanks again for your input.

---

### Official Review · Reviewer_me7c · 2024-11-04

**Soundness:** 3
**Presentation:** 3
**Contribution:** 3
**Rating:** 6
**Confidence:** 4

**Summary:**

The paper presents both feedforward (FF) and recurrent neural network (RNN) models that are trained with the same self-supervised loss function but different inputs such that the RNN but not the FF model learns to path integrate. With an appropriate choice of loss function hyperparameters, both the RNN and FF models develop units with grid-like spatial representations, supporting a recently proposed hypothesis that the function of grid cells is not to support path integration. However, only the RNN develops band-like units. The authors present evidence that it is the band-like units, not grid units, that primarily support the RNN’s ability to path integrate.

**Strengths:**

The authors adapt and simplify a self-supervised loss function that leads RNNs to develop grid-like representations. The authors also show evidence in support of a novel hypothesis that it is band-like units and not grid-like units that are primarily driving the RNN’s ability to path integrate. This suggests a new interpretation for neurons with these response profiles in the brain.

**Weaknesses:**

There are many design choices that influence the representations that RNNs develop. My primary concern is that the conclusions from this RNN model are idiosyncratic to it and do not actually teach us anything about the brain, or perhaps even about RNN models more generally. For example, the authors mention other models that accomplish path integration using RNNs with purely grid-like representations. So we are left with the conclusion that in some models grid-like units are responsible for path integration and in some models, like this one, they are not. This being said, I appreciate the novel modeling work in this study and the new evidence in support of the interesting hypothesis raised by Schoyen et al. iScience (2023) that band-like units, and not grid-like ones, serve a primary role in path integration performance.

**Questions:**

* The RNN in this model path integrates for 10 timesteps. This seems small relative to some other models. For example, I believe Schaeffer, Khona, et al. 2023 used 60 timesteps, Banino et al. 2018 used 100 timesteps, and Cueva & Wei 2018 used 500 timesteps. This gets back to the relevance of the model for biology. The ratemaps produced for a neuron in an animal are taken from long trajectories where the animal traverses large portions of the environment. What do the ratemaps look like if you let your model run for more timesteps and gather unit activity from a single long trajectory? Do you still see grid-like and band-like patterns?

* The finding that a feedforward network can develop grid-like responses without path integration was also observed by Dordek et al. eLife 2016 so this should be cited.

* Do any of the feedforward models, that do not path integrate, have band-like units? For example, the Grid Score of the FF model in Figure 1d also has a peak near 0 (although less pronounced than the RNN), and some of the FF models in Figure A1 have Grid Scores that peak near 0, potentially indicating the presence of band-like units.

* How well can the RNN be used to path integrate? For example, what is the difference between the ground truth (x,y) trajectory specified by the velocity inputs, in comparison to the (x,y) estimate if you linearly combine the activities of the recurrent units (g) by training a matrix of size 2 x 256 to weight the 256 RNN units?

* Presumably the RNN does not have band-like representations before the RNN is trained. Does the RNN develop these band-like representations at the same time it starts to perform well on the path integration task? Or does the RNN perform well on path integration before these band-like representations have developed?

* Do the other models you trained with different hyperparameters also rely on band-like units for path integration? In other words, if a model doesn’t have units with Grid Scores near 0 does this mean it cannot path integrate?

---

> ### Author Response · Authors · 2024-11-23
> **Reply**
>
> Thank you for your insightful comments. While it is true that other RNN models perform path integration using only grid-like units, it is also the case that these models use a non-standard recurrent architecture, wherein the recurrent weight matrix is itself a neural network, dependent on the incoming velocity input. In contrast, neurons in our network are computed using a standard weighted summation of velocity and recurrent inputs. We hold that, while neither model is particularly biologically plausible, our model is simpler, and at least closer to standard models of recurrent models of neuron information integration. We will update our discussion to reflect this position.
>
> On the usage of short trajectories: We train on short sequences to simplify training (shorter sequences allow for more parallelized computation making these models faster to train) and mitigate vanishing/exploding gradients. Additionally, in its simplest form, path integration is a Markovian process, and so only short trajectories should be required to learn perfect path integration. Whether this is biologically plausible is, on the other hand, a fair critique. However, we did evaluate our model on long sequences (shown in Fig. 3c)), which demonstrate that grid-like representations persist for long sequences, even generalizing outside the training environment. We realize, however, that this does not come across clearly from either the main text or caption, so we will update both.
>
> On Dordekt et al.: thank you for raising this point; we will refer to Dordek et al. in the text. The major difference between our work and theirs, is that they do PCA on simulated place cell representations, learning that grid cells can act as a low-dimensional representation of place cells. Our (feedforward) model, on the other hand, and the representations learned by it are solely defined by the distance-preservation/capacity task.
>
> Regarding the possibility of other unit types in the feedforward network: The answer is no: While some units exhibit less pronounced hexagonality, they all do so to some extent. We can, for instance, include ratemaps of every network unit for both networks, in a supplementary figure.
>
> On path integration ability: While we use the state distance (between pruned and unpruned networks) to gauge path integration ability in a decoder-agnostic way, we realize that we do not clearly demonstrate that the RNN is capable of path integration, beyond demonstrating that the network produces consistent ratemaps (for which path integration is required). We will therefore train a linear decoder (as we have observed that linear decoders sometimes struggle to accurately decode locations), and include a population decoding strategy, and demonstrate this capability explicitly - thank you for raising this point.
>
> On band emergence: Correct - the RNN is initialized using random weights, and the resulting initial representations are indeed random. We will add a supplementary analysis of path integration ability as a function of training time, and correlate this with band/grid scores of network units.
>
> On non-grid networks: Again, this is an excellent point. We will train additional networks with lower/higher capacity (which strongly influences learned representations) and see whether band-like representations are present, as this might point to bands as a general mechanism for path integration.

---

> ### Author Response · Authors · 2024-11-27
> **Revision**
>
> Thank you for your patience. In the current revised version, we include several analyses (appendices) that we hope will answer your questions, including
> - evaluating the model on long sequences (Appendix A7)
> - Training a modified linear decoder to gauge decodability (Appendix A7)
> - Included Dordek et al. in a comparison of our model with others (Appendix A4)
> - included full ratemaps of all units in RNN and FF networks, demonstrating that all units are grid-like (Appendix A6)
> - Included a band-analysis as a function of training time, demonstrating that more strongly band-like representation correlate with increased path integration ability during training.  (Appendix A7)
> - Train a low-capacity RNN, and demonstrate that it does not have clear band-like representations, while still learning consistent spatial representations (which requires path integration). (Appendix A6)
>
> We hope this answers your most pressing questions, thanks again for your input.

---

> ### Comment · Reviewer_me7c · 2024-12-03
>
> I appreciate the effort you have made to incorporate feedback and update your paper. The experiments in Figure A4 are quite interesting where you vary the speed and observe more band-like units and even square patterns with a potential connection to Cueva & Wei 2018. This is a nice example (along with the other new figures and Table 1) where I felt like your main modelling results are put into a larger context. In general, this context is exactly what I appreciate and want more of to 1) make connections between the many RNN models of grid cells, and 2) show how your models fit into the larger solution space of models (that may or may not have the grid and band-like properties you are searching for).
>
> I am still curious if I should interpret band-like units as only one potential mechanism for path integration in a standard RNN, or if I should think of this as the primary mechanism. In Figure A7 (great addition!) you have demonstrated a nice correlation between band score and path integration ability. However, can other RNNs like the ones in Figures A6 and A8 for example, without prominent band-like units, path integrate to a similar extent?
>
> Thank you again for your engagement. I enjoyed your revised paper (as well as the initial submission) and found it thought provoking.

---

> > ### Author Response · Authors · 2024-12-03
> >
> > Thank you for insightful input, and we're glad you enjoyed the added analyses. While we unfortunately didn't have the time to perform a full comparison between grid- and non-grid models in terms of path integration ability, we would be happy to do so in a camera-ready version, if time allows. However, we hope that the ratemaps of the low-capacity RNN model demonstrates that such models are indeed capable of path integration (as can be seen from their consistent ratemaps, which are computed from multiple trajectories and visits to a given location), and that bands do not appear needed in the non-grid case.
> >
> > We do believe that bands are the primary mechanism of path integration in standard RNNs, *when* the representations are grid-like, given the range of different analyses that we have performed that all point toward the same conclusion. However, that is not to say that other mechanisms are impossible, but it seems likely that these are either difficult to arrive at through standard optimization, or less optimal than their banded counterparts.
> >
> > Thanks again for helpful discussion.

---

### Official Review · Reviewer_KjQh · 2024-11-04

**Soundness:** 3
**Presentation:** 3
**Contribution:** 2
**Rating:** 5
**Confidence:** 4

**Summary:**

This model studies computational models of grid cells. The authors constructed and investigated two classes of models: Feedforward models (FF) and Recurrent Neural networks (RNNs). They found that under certain training objectives, both classes of models led to hexagonal grid-like firing patterns. Their training objective encourages the network representation to preserve the local distance structure. The paper also performed several additional analyses to understand the trained networks. Some of the results are nice and interesting, although overall the framework is a bit incremental.

**Strengths:**

Originality: This study follows a line of work using machine learning techniques to understand the grid cells (e.g., Cueva&Wei, 2018, Banino et al, 2018, Sorscher et al, 2022, Whittington et al, 2020, Xu et al, 2022. ) The exact learning objective (Eq. 1) seems to be new, but similar objective functions have been proposed in the literature.

Quality: The authors studied and compared two classes of models. They also performed several additional analyses to understand the trained networks, for example,  the connectivity analysis, the analysis on band cells,  and the generalization of the firing pattern outside the training box. These are strengths of the paper.

Clarity: The presentation is generally clear, except for a few places (see below).

Significance: The paper tries to determine the exact ingredients that would lead to grid-firing patterns in the trained neural networks. This seems to be a step toward the right direction.

**Weaknesses:**

I feel that some of the results in the paper are quite interesting, yet the work is a bit incremental. The constraint to preserve local physical distances in the representational space appears to be similar to the conformal isometry hypothesis proposed by Xu et al, 2022, and further investigated in Xu et al, 2024. In the current study, a Gaussian envelope is used to specify the spatial scale that the distance-preserving constraints should be enforces. A discussion on the similarity and difference of the model here and those proposed by Xu et al would be useful for the readers.

The author emphasized the L_1 activity-based regularization in Section 2. However, the consequence of it was not clearly described in the paper. It would be helpful if the authors could show some results to compare the learned firing patterns based on L_1 v.s. L_2 regularizations.


The band cell were also reported in some prior modeling work, e.g., Cueva and Wei 2018. There is some experimental evidence (although slightly controversial) on band cells in medial entorhinal cortex. It would be nice to refer to these results.

The model was not as clearly described. For example, it is not exactly clear what were the inputs to the hidden layers in the FF model. In the figure, it seems that only (x,y) was the input, but I didn’t find a description in the text or Method section.

The paper would be strengthened if the effects of the key hyper-parameters were reported. These include the spatial envelope scale parameter and the weight for the first term in the loss function.

**Questions:**

Does the L_1 regularization (as opposed to L_2 regularization) in the loss function really matter in terms of the grid patterns?


I don’t fully understand the title “grid cells without path integration”. For the RNNs considered in the paper, there was a velocity input. In what sense this is without path integration? I saw the statement that the grid-like cells in the models “appear substantially less important than band cells for path integration in the recurrent network”, but I don’t think that’s equivalent to say that grid cells show up without path integration in the RNNs. If removing the velocity inputs still result in grid firing patterns in RNNs, that would be justified. I’d appreciate if the authors could further clarify this point.


The authors report binomial grid cell orientations in some model variants. Were these results consistent with the neural data?

The units in Fig. 1d is not specified. In particular, what are the units for orientation and spacing?

---

> ### Author Response · Authors · 2024-11-23
> **Reply**
>
> In general, we would like to thank you for your helpful feedback. Regarding the comparison between our work and related models, including those by Xu et al. and Schaeffer et al., we will include an expanded discussion in our manuscript. This discussion will directly compare our approach to these models, highlighting both similarities and differences to provide a clearer context for our contributions.
> We will also add a discussion and study on L1 vs. L2 capacity constraints, which also adds to the discussion on our model vs. others.
> Also, we will add a supplementary figure that shows learned representations as a function of regularization norm.
>
> On band cells: We will refer to cueva et al., and refer explicitly to experimental result concerning band cells. For transparency, we will also make sure to refer to the controversy surrounding band cells (e.g. Navratilova et al., 2016; Grids from bands, or bands from grids? An examination of the effects of single unit contamination on grid cell firing fields)
>
> Regarding the description of the inputs to the model, this is described at the start of the methods section. We acknowledge that this could be made clearer - thank you for pointing this out. To improve clarity, we have rewritten it, and included standard mathematical notation to supplement the textual description of the inputs to each network.
>
> Regarding effects of hyperparameters: We understand that the importance of each hyperparameter should be presented more clearly. We do perform a hyperparameter search in terms of the loss weighting parameter alpha, and the Gaussian tuning scale sigma, located in the appendix. Although this was mentioned in Section 3.2, we had not included a reference to the appendix. This will be included in the updated manuscript and discussed more clearly in the revised version.
>
> Concerning the title: This is a great point, and the reason we use the wording “without path integration” is two-fold: First, we find that grid-like representations emerge in feedforward networks that do not path integrate. Second, in recurrent networks, we find that we can remove velocity input to grid cells, with minimal impact on path integration performance. However, we agree that band-type units could still arise from the architectural difference between networks. We have therefore trained recurrent networks on datasets with different velocities, ranging from 0 velocity, to five times that used in the other models used in this work. Notably, band units do not emerge in the RNN that is trained with zero velocity input. We will add this figure to the supplementary, and update the results section to reflect this - thank you.
>
> On orientations of grid units: It is true that we observe a range of orientations in the feedforward network, and that there seems to be two dominant orientations, which could be consistent with modularity in biological grid cells (see e.g. Stensola et al., 2012, The entorhinal grid map is discretized). However, for the RNN, we observe only a single orientation and module, with minor variations, which is seen in biological cells (see e.g. Redman et al. 2024, Robust variability of grid cell properties within individual grid modules enhances encoding of local space). We will discuss the discrepancy between models in more detail in the updated text.
>
> Concerning Fig. 1d: Thank you for pointing this out. As results are purely simulated, units of grid spacing are arbitrary, and relative to the environment size (described in the methods section), while the orientation is given in radians. We will update the figure to reflect this.

---

> > ### Comment · Reviewer_KjQh · 2024-11-27
> >
> > Thank you for your response. The response clarify some issues.
> >
> > Two questions that I hope to get futher clarification so I can more confidently judge the contribution of the paper:
> >
> > (1) Re:  L_1 v.s. L_2 regularization
> > Are the results qualtatively different when using different types of regularization?  (In the paper, the author emphasized the L_1 activity-based regularization as being a prominent feature of the present model.)
> >
> > (2) Re: the difference to Xu et al. and Schaeffer et al.
> > Can the authors summarize the key differences?

---

> ### Author Response · Authors · 2024-11-27
> **Update**
>
> We have now added several appendices, that we hope will meet your concerns, including
> - A substantial discussion on our work in relation to other models (appendix A4)
> - Demonstrations of L2 capacity constraints (Appendix A8)
> - Made improvements to text and citations for legibility
> - We have updated the model and hyperparameter search description
> - We have added an analysis of the scale parameter sigma to showcase its contribution (and trained models with other alpha values) (Appendix A9)
> - We have made trained models with varying speed distributions. Notably, for no velocity, band-like representations vanish in the RNN. Conversely, for large velocities, bands become more numerous.  (Appendix A5)
>
> We hope this answers your questions, please reach out if you have other concerns.

---

### Author Response · Authors · 2024-11-23
**Revision**

We would like to thank all the reviewers for their time and contributions. We will revise the text throughout the week, adding figures as soon as we can. Please reach out to us if there are additional experiments you would like to see, and we will strive to perform them, within time constraints.

*Update*: We have now posted a first revision of the manuscript, which contains a range of new analyses (specifics in comments). This version still requires some proofreading, which we will continue to do over the course of the day.

---

### Meta-Review · Area_Chair_PaE6 · 2024-12-22

**Metareview:**

This submission studies emergence of grid-like and band-like representations in neural networks using a self-supervised objective based on local distance preservation and capacity constraints. The authors demonstrate that grid cells emerge in both feedforward and recurrent networks, with band cells playing a central role in path integration. The work challenges the traditional understanding of grid cells as essential for path integration, proposing instead that their emergence is tied to local spatial encoding.

Overall, while the paper is interesting and competently executed, the incremental nature of its contributions and limited biological alignment reduce its suitability for a selective venue like ICLR. See below for details.

**Additional Comments On Reviewer Discussion:**

Reviewers agreed that the study is supported by thorough experiments. The results on band cells were a valuable contribution. However, the reviewers noted that the paper's contributions are incremental relative to prior work, such as Xu et al. and Schaeffer et al. The novelty of the loss function is limited, and its biological plausibility is questioned, particularly given the lack of neuron type diversity and multi-modular grid cells seen in biological systems. Furthermore, while the authors addressed some of these issues in revisions, the comparison to related work remains insufficiently highlighted in the main text.

---

### Decision · Program_Chairs · 2025-01-22

Reject